# In vitro induction of patterned branchial arch-like aggregate from human pluripotent stem cells

Yusuke Seto [1,2] ✉, Ryoma Ogihara[2], Kaori Takizawa[1] & Mototsugu Eiraku [1,2,3] ✉

Early patterning of neural crest cells (NCCs) in the craniofacial primordium is important for subsequent development of proper craniofacial structures. However, because of the complexity of the environment of developing tissues, surveying the early specification and patterning of NCCs is difficult. In this study, we develop a simplified in vitro 3D model using human pluripotent stem cells to analyze the early stages of facial development. In this model, cranial NCC-like cells spontaneously differentiate from neural plate border-like cells into maxillary arch-like mesenchyme after a long-term culture. Upon the addition of EDN1 and BMP4, these aggregates are converted into a mandibular arch-like state. Furthermore, temporary treatment with EDN1 and BMP4 induces the formation of spatially separated domains expressing mandibular and maxillary arch markers within a single aggregate. These results suggest that this in vitro model is useful for determining the mechanisms underlying cell fate specification and patterning during early facial development.

Facial tissues are essential for animal survival; hence, the cellular mechanisms involved in face formation during development are important. The ectoderm-derived cell population called neural crest cells (NCCs) contributes to most of its structures[1]. NCCs are transient cell populations that arise from the neural plate border (NPB). They migrate from their dorsal origin to the ventral craniofacial primordium and contribute to the cartilage, bones, and connective tissues.

The facial primordium is divided into three parts, the frontonasal prominence, the maxillary arch (MX), and the mandibular arch (MN), which give rise to distinct parts of the face[2]. As studied extensively in the MN, NCCs are further patterned into several domains characterized by the expression of specific genes in the primordium[3]. Stepwise patterning events in the early phase are important for proper facial development. However, the underlying cellular processes are not yet fully understood. Environmental signals in which NCCs migrate are thought to be strong determinants of their fate; however, a recent study revealed that the movements of postmigratory NCCs within the mandibular arch are involved in cell fate specification through sequential fate bifurcations[3]. This study suggests that early-stage

patterning of NCCs is a more complicated process than previously thought. Due to technical limitations and the complexity of the environment in developing tissues, there are some difficulties in studying such early processes in vivo. Therefore, we aimed to establish an in vitro model that recapitulates these early processes during facial development.

Human pluripotent stem cells (hPSCs), such as embryonic stem cells (ESCs) and induced pluripotent stem cells (iPSC), have been used for NCC induction in vitro. In the directed differentiation of NCCs from hPSCs[4–7], balanced WNT/bone morphogenic protein (BMP) signals are important for their induction, similar to NCC induction in model animals[8]. These methods require extrinsic manipulation of signals through the addition of small-molecule regulators. In this study, we established a method to induce a large number of NCC-like cells in a floating culture of hPSC-derived aggregates without additives. NCC induction depends on aggregate-intrinsic WNT/BMP signals. These NCC-like cells were generated from an NPB-like state, and they expressed epithelial-mesenchymal transition (EMT)-related genes during differentiation, indicating that they recapitulated the natural

[1]Laboratory of Developmental Systems, Institute for Life and Medical Sciences, Kyoto University, 53 Shogoin Kawaharacho, Sakyo-ku, Kyoto 606-8507, Japan. [2]Department of Polymer Chemistry, Graduate School of Engineering, Kyoto University, Kyoto 615-8510, Japan. [3]Institute for Advanced Study of Human Biology (WPI-ASHBi), Kyoto University, Kyoto, Japan. ✉e-mail: seto.yusuke.gm@gmail.com; eiraku@infront.kyoto-u.ac.jp

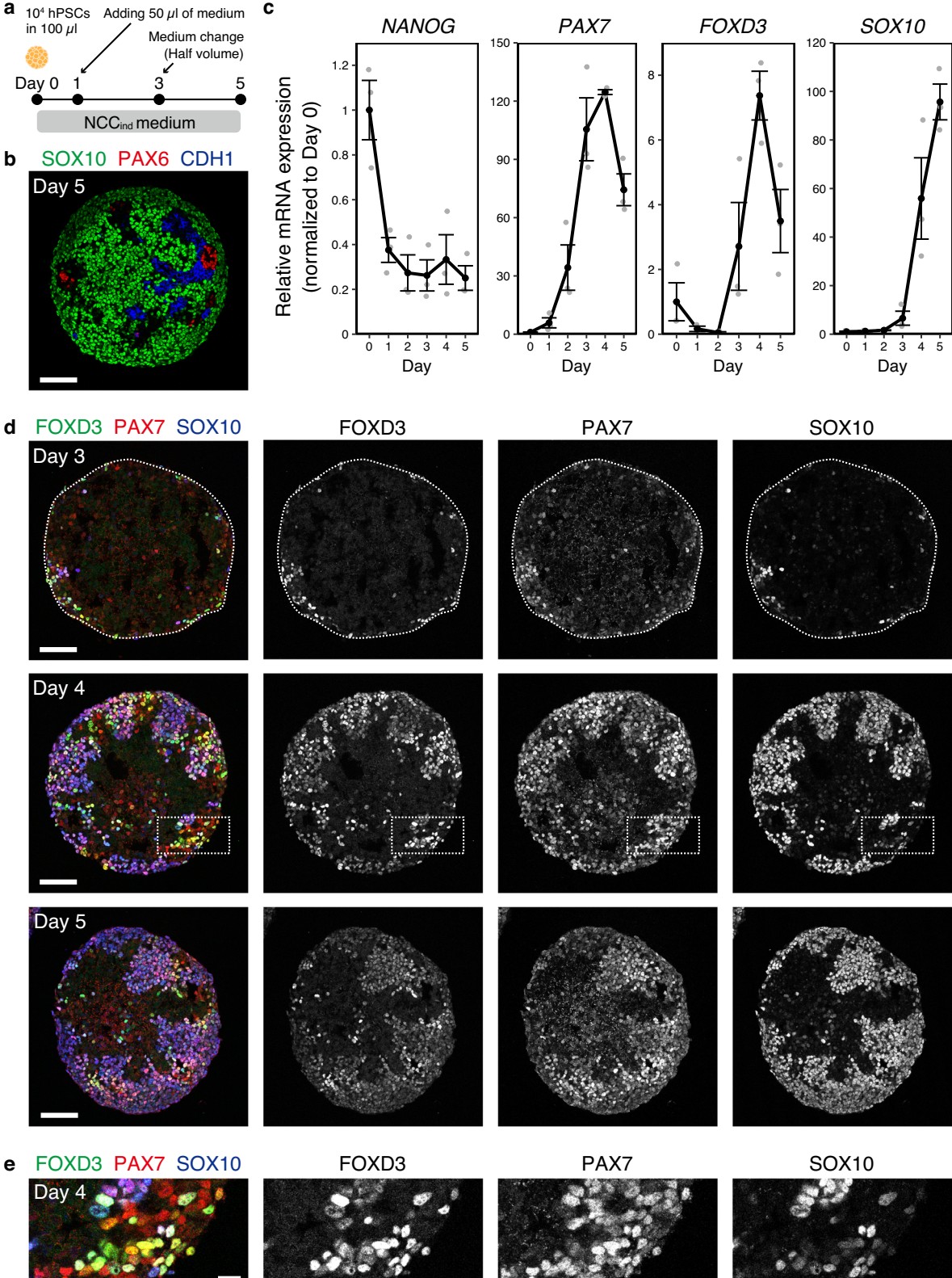

**Fig. 1 | Induction of NCC (neural crest cell)-like cells in the aggregates derived from hPSCs (human pluripotent stem cells). a** Scheme of the differentiation culture for the induction of NCC-like cells. **b** Immunostaining of the day-5 aggregate with antibodies for SOX10 (green), PAX6 (red), and CDH1 (blue). Scale bar, 100 μm. Three independent cultures were used for experiment and representative images are shown. **c** Real-time polymerase chain reaction analysis of time-course samples. Day-0 samples were undifferentiated hPSCs and used for normalization. Data are presented as mean ± standard error of mean (*n* = 3 independent experiments). Source data are provided as a Source Data file. **d** Immunostaining of the aggregates at days 3, 4, and 5 with antibodies for FOXD3 (green), PAX7 (red), and SOX10 (blue). Scale bar, 100 μm. Three independent cultures were used for experiment and representative images are shown. **e** Magnifications of areas surrounded by dotted lines in images of day-4 aggregates in (**d**). Scale bar, 20 μm.

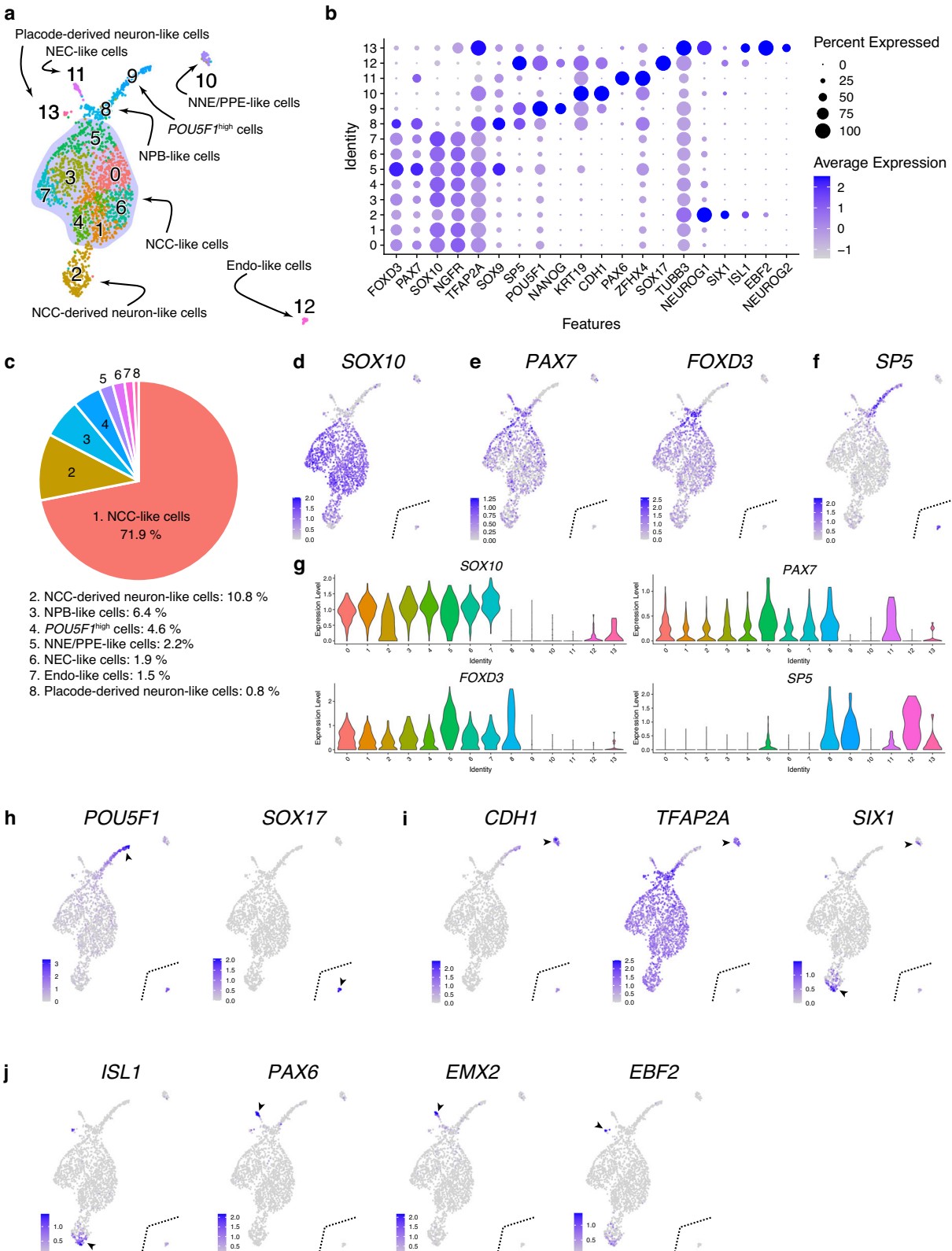

Fig. 2 | **Analysis of the cellular content of the day-5 aggregates. a** Uniform manifold approximation and projection of scRNA-seq data from the day-5 aggregates. Cluster number was determined according to the cluster size. Endo endoderm, NCC neural crest cell, NEC neuroepithelial cell, NNE/PPE non-neural ectoderm/pre-placodal epithelium, NPB neural plate border. **b** Dot plot of key marker genes to define each cluster. **c** Ratio of the annotated populations in the day-5 aggregates. **d–f** Feature plots of genes which characterize NCC-like cells and their progenitors **g** Violin plots of *SOX10*, *PAX7*, *FOXD3*, and *SP5* (**h–j**) Feature plots of genes characterize immature cells, endo-like cells, NNE/PPE-like cells, NCC-derived neuron-like cells, NEC-like cells and placode-derived neuron-like cells.

developmental processes. We tested the potential of these aggregates as an in vitro model of early facial development using a long-term culture. Long-term culture-induced MX markers, such as *POU3F3* in aggregates[9]. Furthermore, the addition of endothelin 1 (EDN1) and BMP4, secreted from the epithelium surrounding the MN in vivo[10,11], could induce MN markers, such as *HAND2*, and repress MX markers[12]. Temporary treatment with EDN1/BMP4 induced both POU3F3[+] and HAND2[+] regions separately within single aggregates, suggesting intrinsic patterning activity in the aggregates. Prolonged culture of the aggregates also permitted the differentiation of cells into osteogenic/chondrogenic lineages. These results indicate its usefulness as an in vitro model for early patterning and cell fate specification in the facial primordium.

## Results

### Spontaneous induction of NCC-like cells in hPSC aggregates

To induce NCCs from hPSC, we cultured hPSC aggregates in a defined medium (NCC$_{ind}$ medium) (Fig. 1a). Day-5 aggregates were enriched in SOX10[+] NCC-like cells. However, they contained a small number of PAX6[+] neuroepithelial cell (NEC) and CDH1[+] non-neural epithelium (NNE) cells (Fig. 1b). Knockout serum replacement (KSR), which is often used in neural tissue induction, reduced the induction of NCC-like cells (Supplementary Fig. 1)[13,14]. Therefore, we added B-27 supplement instead of KSR to improve cell viability and we analyzed the gene expression during differentiation (Fig. 1c). The hPSC marker *NANOG* was downregulated on the first day. One of the earliest NCC markers, *PAX7*, was also upregulated. Subsequently, the early NCC marker *FOXD3* and the late marker *SOX10* were upregulated. We confirmed the expression of these genes using immunohistochemistry (IHC) (Fig. 1d). Under our experimental conditions, we detected cells co-expressing all these markers in the peripheral region of the aggregates from day 3. The expression of FOXD3 and SOX10 mostly overlapped at every stage tested, although the expression of FOXD3 seemed to be weaker on day 5 than on day 4. In model animals, such as chicks and zebrafish, FOXD3 and SOX10 are expressed in distinct but partially overlapping patterns, and FOXD3 seems to be downregulated as development proceeds[15,16]. Weaker expression of FOXD3 at day 5 may reflect the downregulation observed in vivo. The expression of PAX7 was broader than that of the others, especially on day 4 (Fig. 1d, e). FOXD3[-]/SOX10[-]/PAX7[+] cells might be earlier NCC precursor-like cells or other types of cells, such as dorsal NECs[17].

As day-5 aggregates contained non-NCC-type cells (Fig. 1b), we analyzed the content of aggregates derived from KhES-1 cells by using single-cell RNA-seq (scRNA-seq). A total of 1877 cells were divided into 14 clusters using unsupervised uniform manifold approximation and projection (UMAP) clustering (Fig. 2a, b). Clusters 0, 1, 3, 4, 5, 6, and 7 represented *SOX10*[+] NCC-like cells, accounting for 71.9% of the total cells (Fig. 2c, d). The expression of *PAX7* and *FOXD3* was higher in cluster 5 than in the NCC-like clusters, indicating that cluster 5 represented nascent NCC-like cells (Fig. 2e, g). The presence of PAX7[high]/FOXD3[high] cells was confirmed using IHC (Fig. 1d). Cluster 8, adjacent to cluster 5, represented *SP5*[+]/*PAX7*[+]/*FOXD3*[+] NPB-like cells (Fig. 2f, g)[18]. Cluster 8 was also characterized by extremely low SOX10 expression (Fig. 2b, d, g). Cluster 9 represented more immature cells, characterized by higher *POU5F1* expression and the absence of *SOX17* expression, a marker gene for endodermal cells (Endo) (Fig. 2h)[19]. Cluster 10 contained *CDH1*[+]/*TFAP2A*[+] NNE-like cells, which also contained *SIX1*[+] pre-placodal epithelium (PPE)-like cells (Fig. 2i)[20]. Three types of neural lineage clusters were observed (Fig. 2j). Cluster 2 appeared to represent *ISL1*[+]/*SIX1*[+] NCC-derived neuron-like cells (Fig. 2i, j)[21] and cluster 11 contained *PAX6*/*EMX2*[+] NEC-like cells (Fig. 2j)[22]. Cluster 13 comprised *EBF2*[+] placode-derived neuron-like cells (Fig. 2j)[23]. Cluster 12 represented *SOX17*[+] endo-like cells (Fig.2h). We also confirmed the presence of these cell types using IHC (Supplementary Fig. 2).

Although p75 is not specific to NCCs in human embryos[24], scRNA-seq revealed that p75 expression was mostly specific to NCC-like cells in day-5 aggregates (Supplementary Fig. 3a). IHC also showed that p75 expression was specific to SOX10[+] cells (Supplementary Fig. 3b, c). Almost consistent with scRNA-seq data, 76.5% of the cells in day-5 aggregates derived from KhES-1 expressed p75 at a high level on average (Supplementary Fig. 3d). Although the induction efficiency of SOX10[+] cells varied among the hPSC lines, the expression of p75 appeared to be specific to SOX10[+] cells in all cell lines (Supplementary Fig. 3b–d). These results indicate that p75 could be a suitable surface antigen for isolating NCC-like cells in this experimental system. We also tested the multipotency of these NCC-like cells (Supplementary Fig. 4). These cells can differentiate into neuronal cells, glial cells, melanoblasts, and ectomesenchymal derivatives, including smooth muscle cells, osteoblasts, chondroblasts, and adipocytes.

In our method, NCC-like cells were spontaneously induced without additives, such as CHIR-99021, BMPs, and SB-431542 (SB) that are often used in other methods[4–7], indicating the importance of aggregate-intrinsic signals. We analyzed the role of aggregate-intrinsic WNT/BMP signals, considering their importance in vivo (Fig. 3a)[8]. *WNT1/4* were expressed in aggregates and peaked on days 3 and 4. *AXIN2*, a well-known downstream target of the canonical WNT signal[25], also peaked around days 3 and 4. A non-canonical WNT, *WNT7B*, was upregulated on day 1. *BMP2* was also upregulated around day 3, whereas *BMP4* was upregulated between days 0 and 1, suggesting that *BMP4* may play a role in the early phase of differentiation. To reveal the function of WNT/BMP signals in our method, we treated the aggregates with the WNT inhibitor IWP-2 or the BMP inhibitor dorsomorphin (Fig. 3b). We added these inhibitors from days 1, 2, or 3 to day 5 and analyzed the expression of marker genes for NCC-like cells (*SOX10*), NEC-like cells (*PAX6*), NNE-like cells (*CDH1*), and PPE-like cells (*EYA1*) (Fig. 3c). WNT inhibition starting on day 1 strongly suppressed *SOX10* (Fig. 3c). This effect was weakened when IWP-2 was added on day 2. Treatment from day 3 showed almost no effect, suggesting that WNT signaling before day 3 was important for NCC-like cell induction. However, canonical WNT activity was upregulated from day 3, suggesting the importance of the early expression of non-canonical WNTs, such as *WNT7B*. BMP inhibition on day 1 also suppressed *SOX10* expression, although this suppression was incomplete. This indicates that BMPs, such as BMP4, might act from day 0, and that their activities were sufficient to induce the differentiation of subsets of NCC-like cells. BMP inhibition on day 3 was also ineffective on *SOX10* expression. For *PAX6*, *CDH1*, and *EYA1*, both WNT and BMP inhibition from day 3 had almost no effect, indicating that the fate of most precursors was determined by day 3 (Fig. 3c). WNT inhibition from days 1 and 2 slightly upregulated *CDH1* and *EYA1*, indicating an increase in the number of NNE/PPE-like cells. Conversely, BMP inhibition on day 1 significantly reduced the expression of *CDH1* and *EYA1*, indicating a reduction in the number of NNE/PPE-like cells. BMP inhibition from day 1 also resulted in the upregulation of *PAX6*, indicating an increase in the number of NEC-like cells, which was consistent with a reduction in NNE/PPE marker expression. Interestingly, WNT inhibition from days 1 and 2 also upregulated *PAX6*, although NNE/PPE markers were also upregulated under these conditions. We analyzed the aggregates treated with inhibitors from day 1 using IHC (Fig. 3d, e). Consistent with the results of real-time polymerase chain reaction (PCR), an increase in the number of PAX6[+] cells and a decrease in those of SOX10[+] and CDH1[+] cells were observed in the dorsomorphin-treated aggregates. In contrast, SOX10[+] cells disappeared completely from the IWP-2-treated aggregates (Fig. 3d). In IWP-2-treated aggregates, the counts of PAX6[+] cells were increased, as well as in dorsomorphin-treated aggregates, but these PAX6[+] cells expressed CDH1, suggesting that they were of the NNE/PPE-like lineage. PAX6 is also expressed in the anterior PPE, such as the lens/olfactory placodes[26]. We confirmed

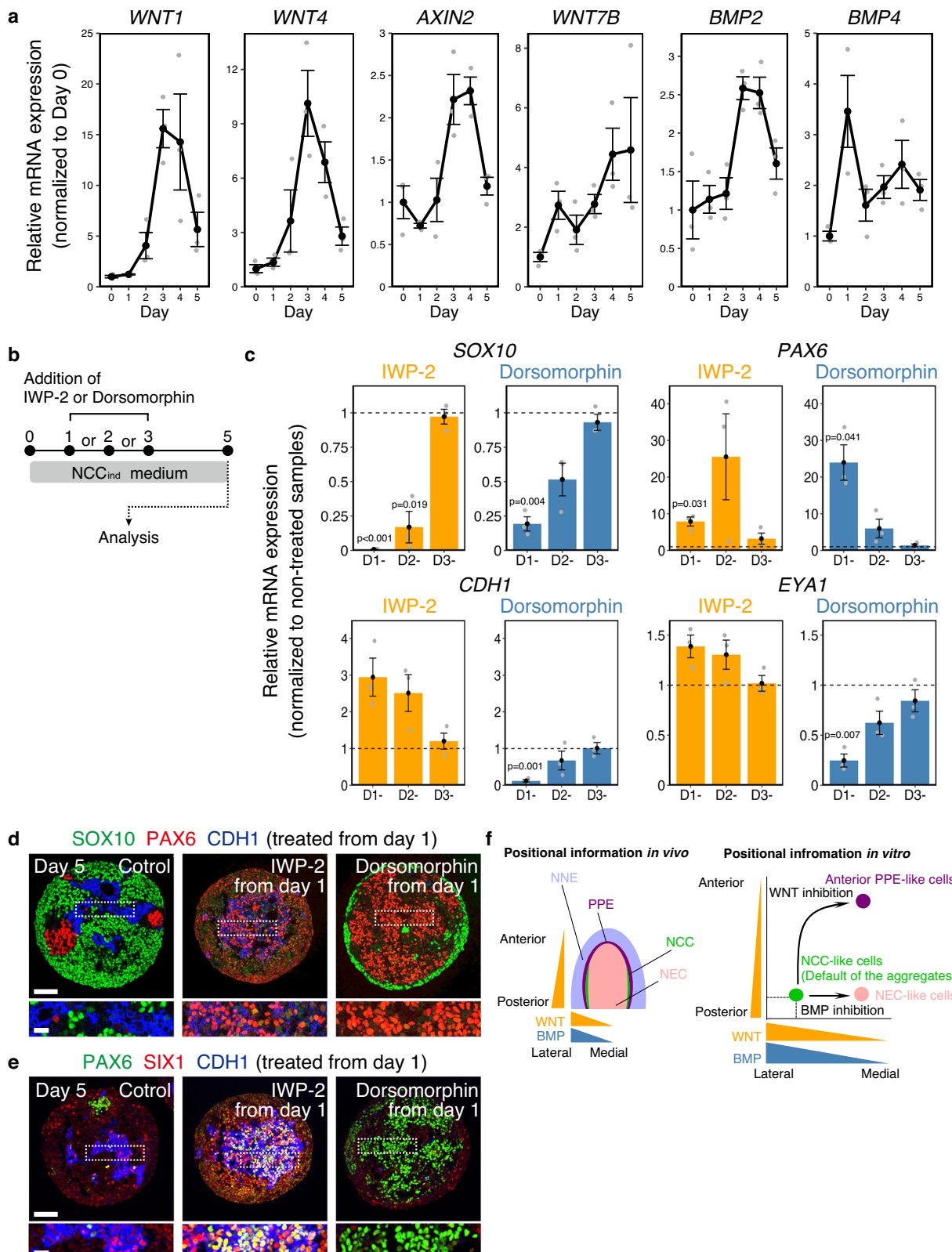

that there were SIX1⁺/CDH1⁺ cells that did not express PAX6 in non-treated aggregates, while many SIX1⁺/CDH1⁺ cells co-expressed PAX6 in IWP-2-treated aggregates, suggesting that they became anterior PPE-like cells (Fig. 3e). Anteriorization by WNT inhibition is consistent with the role of WNT signals in the posteriorization of embryonic ectodermal tissue during early development[27]. For dorsomorphin-treated aggregates, there were hardly any CDH1⁺

NNE/PPE-like cells. These results indicate that aggregate-intrinsic WNT and BMP signals regulate the positional identity of cells to induce NCC-like cells (Fig. 3f).

SB, which antagonizes activin/TGFß signal, was also used in some NCC induction methods[4,6]. We tested the effect of SB on our NCC-like cell induction method by adding SB from day 0 onward. In SB-treated aggregates, the number of NCC-like cells expressing FOXD3 and

**Fig. 3 | Importance of endogenous WNT/BMP (bone morphogenetic protein) signals during NCC (neural crest cell)-like cell induction. a** Real-time polymerase chain reaction (PCR) analysis of *WNT/BMP* and downstream genes in time-course samples. Day-0 samples were undifferentiated hPSCs. Data are presented as mean ± standard error of mean (SEM) (*n* = 3 independent experiments). Source data are provided as a Source Data file. **b** Scheme of the WNT/BMP inhibition experiment. IWP-2 or dorsomorphin was added on days 1, 2, and 3. Aggregates were analyzed by real-time PCR and immunohistochemistry on day 5. **c** Real-time PCR analysis of inhibitor-treated aggregates on day 5. Non-treated aggregates were used as control. Data are presented as mean ± SEM (*n* = 3 independent experiments). Two-tailed Student's *t*-test was used. Source data are provided as a Source Data file. **d** Immunostaining of inhibitor-treated aggregates on day 5 with antibodies for SOX10 (green), PAX6 (red), and CDH1 (blue). Scale bars, 100 μm and 25 μm (magnifications). Three independent cultures were used for experiment and representative images are shown. **e** Immunostaining of inhibitor-treated aggregates on day 5 with antibodies for PAX6 (green), SIX1 (red), and CDH1 (blue). Scale bars, 100 μm and 25 μm (magnifications). Three independent cultures were used for experiment and representative images are shown. **f** Scheme representing a hypothesis about the effect of WNT/BMP inhibition. The left illustration indicates the spatial organization of each cellular domain in the early embryo. WNT and BMP forms gradients along anterior-posterior and medial-lateral axes, regulating positional information in the embryo. The right illustration indicates the hypothetical scheme of the effects of WNT/BMP inhibition in vitro. hPSC-derived aggregates generate NCC-like cells preferentially in NCC$_{ind}$ medium in intrinsic WNT and BMP-dependent manner as default state. WNT inhibition may change the positional information of the aggregate into more anterior and medial identity and result in the induction of the anterior PPE-like cells. BMP inhibition may induce more medial positional identity and result in the induction of neural cells. NEC neuroepithelial cell, NNE non-neural ectoderm, and PPE pre-placodal epithelium.

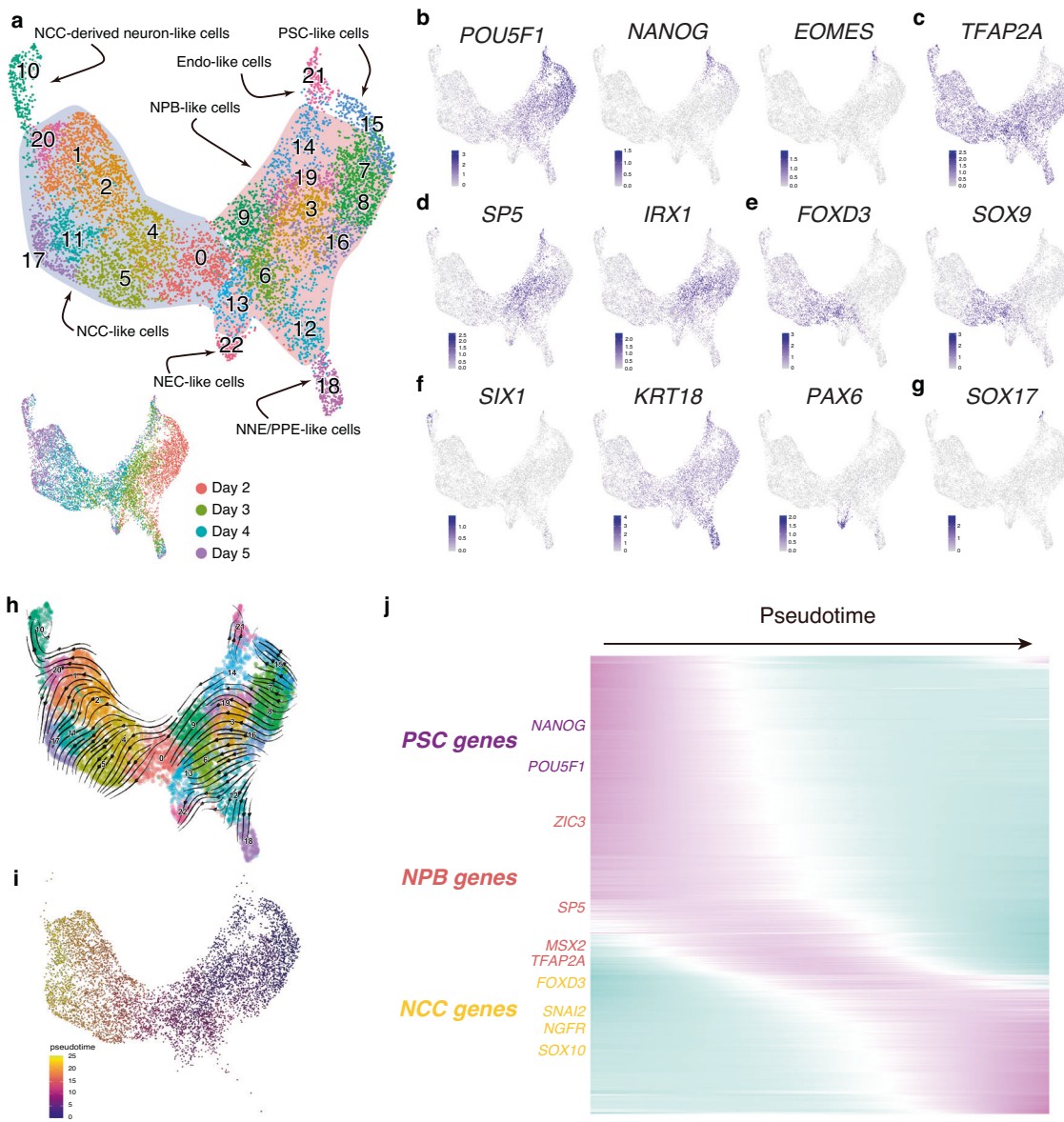

**Fig. 4 | Characterization of the developmental process of the aggregates. a** UMAP (Uniform manifold approximation and projection) of merged scRNA-seq data (day 2 to day 5). Individual cells are colored by cluster number or sampling date. Endo endoderm, NCC neural crest cell, NEC neuroepithelial cell, NNE/PPE non-neural ectoderm/pre-placodal epithelium, NPB neural plate border, PSC pluripotent stem cell. **b**–**g** Feature plots of genes associated with PSC-like cells, endoderm-like cells, NPB-like cells, NNE/PPE-like cells, NEC-like cells. **h** RNA velocities visualized as streamlines on UMAP. **i** UMAP of scRNA-seq data colored by inferred pseudotime without clusters 10, 18, 21, and 22. **j** Expression dynamics of PSC, NPB, and NCC-related genes along pseudotime.

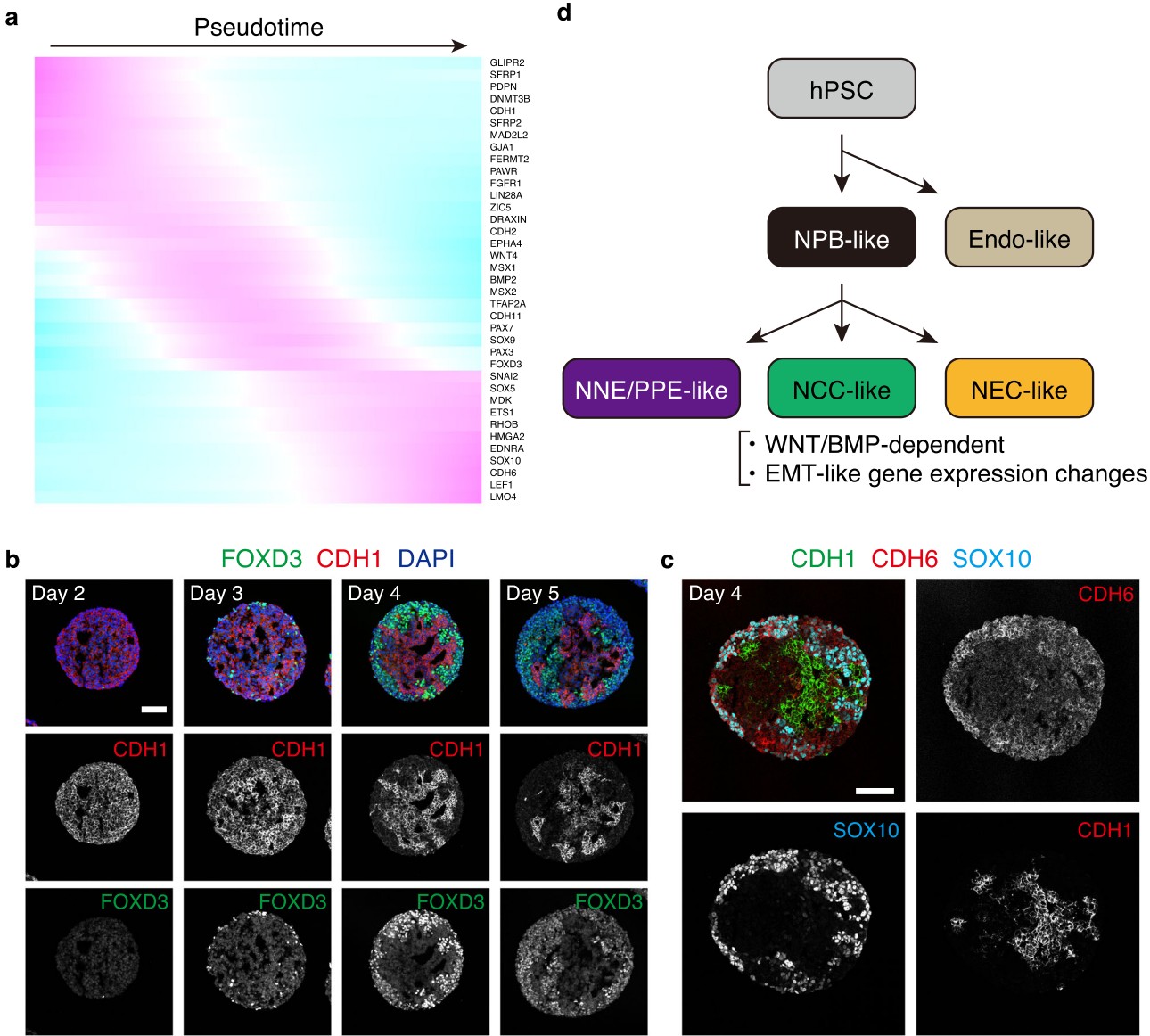

**Fig. 5 | Recapitulation of gene expression changes associated with EMT (epithelial-mesenchymal transition). a** Expression dynamics of EMT-related genes along pseudotime. **b** Immunostaining of aggregates with antibodies for FOXD3 (green) and CDH1 (red), and counterstaining with DAPI (blue). FOXD3 and CDH1 were expressed in complementary manner. Scale bar, 100 μm. Three independent cultures were used for experiment and representative images are shown. **c** Immunostaining of aggregates with antibodies for CDH1 (green), CDH6 (red), and SOX10 (cyan). CDH1 and CDH6 were expressed in complementary manner. SOX10⁺ cells expressed CDH6 instead of CDH1. Scale bar, 100 μm. Three independent cultures were used for experiment and representative images are shown. **d** Schematic representation of the presumptive process of NCC (neural crest cell)-like cell induction from hPSCs (human pluripotent stem cells). hPSCs bifurcate into NPB (neural plate border)-like cells or Endo (endoderm)-like cells at first. Then NPB-like cells differentiate into NNE/PPE (non-neural ectoderm/pre-placodal epithelium)-like cells, NEC (neuroepithelial cell)-like cells, and NCC-like cells. NCC-like cell induction depends on aggregate-intrinsic WNT/BMP (bone morphogenetic protein) signals and is associated with EMT-like gene expression changes.

SOX10 increased, while SOX17⁺ endo-like cells disappeared on day 4 (Supplementary Fig. 5). This result is consistent with the finding that SB inhibits the endodermal induction of hPSCs while facilitating neuroectodermal differentiation[28].

**Recapitulation of in vivo differentiation process during NCC-like cell induction**
We appended the scRNA-seq data of day-2, day-3, and day-4 aggregates to the data of day-5 aggregates for further analysis of the developmental process of the aggregate (total 8466 cells, 23 clusters) (Fig. 4a). The cells in the cluster 15 were *POU5F1⁺/NANOG⁺/EOMES⁻*, indicating that they exhibited a PSC-like state (Fig. 4b)[29]. The

early NPB marker *TFAP2A*, which can induce the expression of NCC marker genes when overexpressed in hPSCs[30], was upregulated in cluster 7 adjacent to cluster 15, and its expression continued throughout differentiation into NCC-like and NNE/PPE-like cells (Fig. 4c). The NPB markers, *SP5* and *IRX1*, were upregulated in groups of clusters adjacent to cluster 7 (clusters 3, 6, 8, 9, 13, 14, 16, and 19) (Fig. 4d)[18,31]. Another group of clusters (clusters 1, 2, 4, 5, 11, 17, and 20) contained *SOX10⁺* NCC-like cells (Supplementary Fig. 6). In this group, some marker genes showed differential expression patterns, suggesting variations in cell state (Supplementary Fig. 6). Cluster 0 was the transition point between the NPB-like and NCC-like clusters expressing *FOXD3* and *SOX9* at high levels (Fig. 4e and

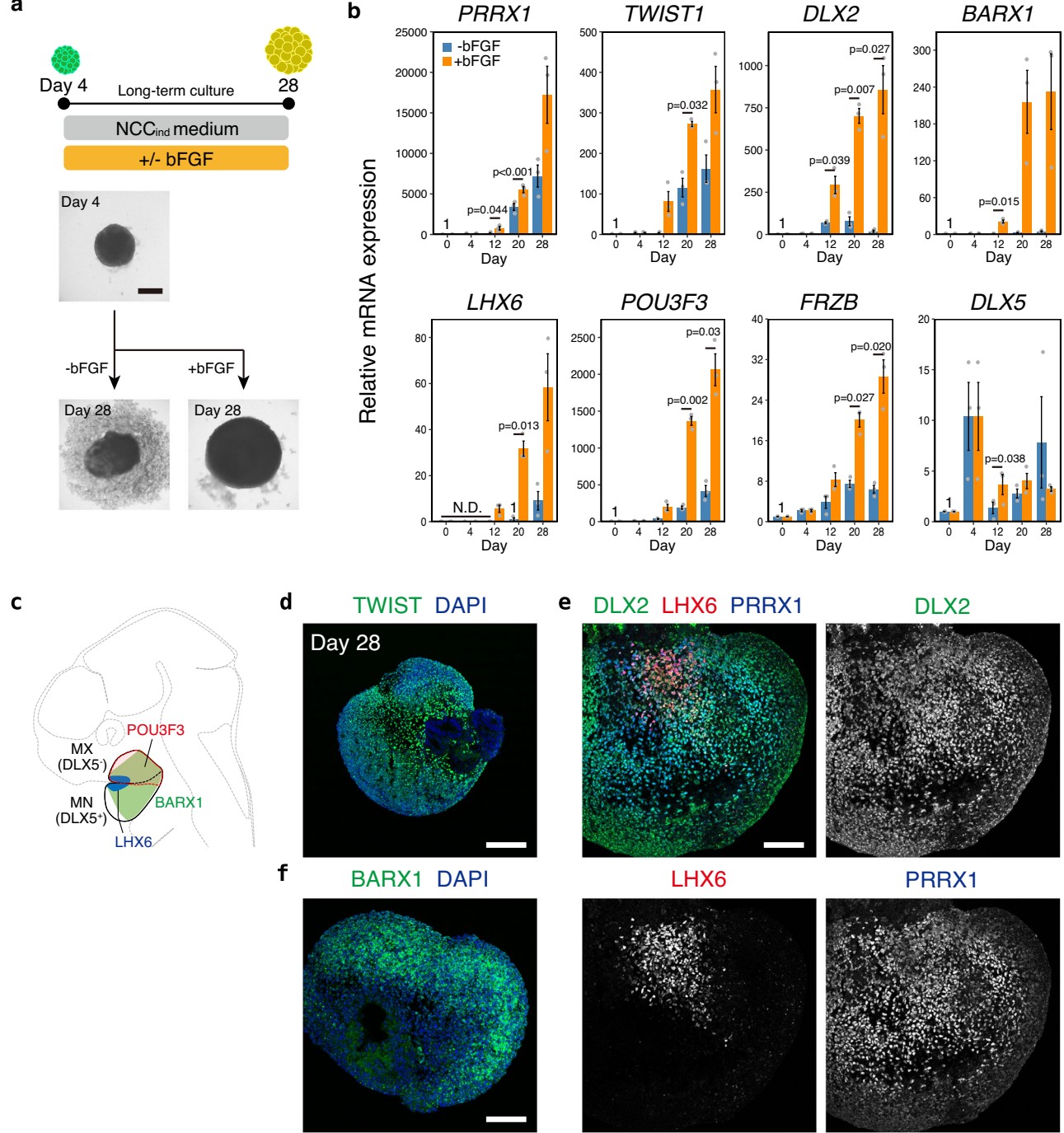

**Fig. 6 | Differentiation of the aggregates into craniofacial mesenchyme-like cells. a** Scheme of the long-term culture and bright-field images of the aggregates. Aggregates were cultured with or without bFGF (basic fibroblast growth factor) from day 4. Scale bar, 400 μm. NCC: neural crest cell. Three independent cultures were used for experiment and representative images are shown. **b** Real-time PCR analysis of craniofacial mesenchyme-related genes. Data are presented as mean ± SEM ($n = 3$ independent experiments). Two-tailed Student's *t*-test was used. Source data are provided as a Source Data file. **c** Schema of gene expression pattern in the first branchial arch. MN mandibular arch, MX maxillary

arch. **d** Immunostaining of bFGF-treated aggregates on day 28 with antibodies for TWIST1 (green) and counterstaining with DAPI (blue). Scale bars, 200 μm. Three independent cultures were used for experiment and representative images are shown. **e** Immunostaining of bFGF-treated aggregates on day 28 with antibodies for DLX2 (green), LHX6 (red), and PRRX1 (blue). Scale bars, 100 μm. Three independent cultures were used for experiment and representative images are shown. **f** Immunostaining of bFGF-treated aggregates on day 28 with antibodies for BARX1 (green) and counterstaining with DAPI (blue). Scale bars, 100 μm. Three independent cultures were used for experiment and representative images are shown.

Supplementary Fig. 6). Cluster 10 contained *SIX1*+ NCC-derived neuron-like cells (Fig. 4f). Clusters 12/18 contained *KRT18*high NNE/PPE-like cells, and cluster 22 contained *PAX6*+ NEC-like cells (Fig. 4f). Cluster 21 contained *SOX17*+ endo-like cells (Fig. 4g). RNA velocity analysis suggested that most of the hPSCs differentiated into NPB-

like states. However, only a small number of cells differentiated into *SOX17*+ endo-like cells (Fig. 4h). NPB-like cells seemed to differentiate into NCC-like-, NEC-like-, and NNE/PPE-like cells, suggesting that NPB-like cells are multipotent, as observed in model animals (Fig. 4h)[32–34].

NPB is located between SOX2⁺ NE and TFAP2A⁺ NNE, and both genes are expressed in model animals[32]. In our scRNA-seq dataset, we observed the co-expression of *SOX2* and *TFAP2A* NPB-like clusters at all stages analyzed (Supplementary Fig. 7a). We also confirmed co-expression in the aggregates using IHC (Supplementary Fig. 7b). Co-expression of SOX2 and TFAP2A clearly appeared on day 2. However, TFAP2A was already expressed in some SOX2⁺ cells weakly at day 1. From day 4, SOX2 expression decreased in most TFAP2A⁺ cells, including NPB-like, NCC-like, and NNE/PPE-like cells. This down-regulation of SOX2 has been observed in the NCCs of chick embryos[32]. Some SOX2⁺/TFAP2A⁺ cells also co-expressed PAX7 from day 3, similar to the NPB cells in chick embryos (Supplementary Fig. 7b).

We analyzed the NPB clusters in more detail to identify the variety in the cell state. NPB clusters (clusters 3, 6, 7, 8, 9, 12, 13, 14, 16, and 19) were extracted according to the expression of *SP5* and *TFAP2A* and re-clustered to identify differentially expressed genes (Supplementary Fig. 8a). *SP5* and *IRX1* were broadly expressed. The cranial NCC (cNCC) markers, *LHX5*, *DMBX1*, and *OTX2*, were also expressed in a large number of cells, although the expression patterns of *DMBX1* and *OTX2* were sparse and restricted, compared with *LHX5*, respectively. *Lhx5* and *Dmbx1* are regulators of the cNCC-specific gene regulatory network in chick embryo[35]. *Otx2* is crucial for head development[36]. These results suggest that most NPB-like cells possess a cNCC-like identity.

The NPB-like cell clusters could be roughly classified into two groups (Supplementary Fig. 8c, d). One group expressed genes related to the anterior patterning of the embryonic ectoderm, such as *CYP26A1* and *HESX1*[37,38]. This group mainly comprised cells of early differentiation days (day 2: 85.0%, day 3: 14.6%, day 3: 0.4%, and day 4: 0%). In the other group, NPB-specified *PAX3* and *MSX1* were broadly expressed instead of *CYP26A1* and *HESX1*. This group comprised cells from late differentiation days (day 2: 13.8%, day 3: 26.9%, day 3: 29.6%, and day 4: 29.7%). This result suggests that early anterior patterning of the ectoderm occurred in the aggregates, followed by further differentiation of NPB-like cells.

We also searched for genes with distinct expression patterns within NPB-like cell clusters, focusing on transcription factors (Supplementary Fig. 9). For example, *FOXD3*, *ZFHX4*, *TFAP2B*, *DACH1*, and *EBF1* are expressed in cells located on the left side of the UMAP. These genes are also expressed in NCCs in vivo[39–42]. *PAX6*, *EN1*, *PAX2*, *PAX7*, *OTX2*, *GBX2*, and *EMX2*, which are essential factors for the patterning of NECs around the diencephalon and mid/hindbrain, also exhibited distinct patterns[22,43], suggesting those factors may define the axial identity of NPB-like cells in the aggregates. *DLX5*, *GRHL2*, *GRHL3*, *MAFB*, and *HAND1* also showed characteristic expression patterns in NPB-like cells, and their expression extended to NNE/PPE-like cell clusters. These genes are involved in the NNE/PPE lineage development in vivo[44–47]. The tumor suppressor *ELF3* is also expressed in similar patterns, although its role in NPB-like cells is unknown[48].

Furthermore, a trajectory analysis of NPB-like cells was performed to identify gene expression patterns specific to the trajectories bound for each ectodermal lineage (Supplementary Fig. 10a). Some genes exhibited differential expression patterns among the different lineages (Supplementary Fig. 10b). These genes included those known to be important for the development of each lineage, as earlier mentioned. There were also several lineage-specific genes whose roles in the differentiation of NPB-like cells were unknown, such as *EBF1* and *SEZ6* for the NCC-like lineage and *ELF3*, *KLF6* for the NNE/PPE-like lineage. For example, *EBF1* is known for its expression in migratory cNCC, but its function in NPB is unknown[41], similar to that of *ELF3*.

To infer the gene expression dynamics during the differentiation of NCC-like cells, pseudotime analysis was performed on clusters, excluding NNE/PPE-like, NEC-like, and endo-like cell clusters (Fig. 4i). Markers for PSC, NPB, and NCC were upregulated along with pseudotime, indicating the recapitulation of the in vivo NCC induction

processes to some extent (Fig. 4j). During in vivo development, EMT is associated with the induction of NCCs as they leave their origin. EMT is characterized by several features, such as the expression of EMT-inducing transcription factors, changes in the expression of cadherin genes, and degradation of the basement membrane[49]. Gene ontology analysis of cluster markers revealed that in marker genes for cluster 0 of the merged scRNA-seq data, which was a junction between NPB and NCC-like cells, were enriched with genes related to EMT (i.e., GO:0001837 "epithelial to mesenchymal transition") (Supplementary Fig. 11). Pseudotime analysis revealed temporal changes in the expression of several genes related to EMT, including transcription factors and cadherins (Fig. 5a). Consistent with their regulatory relationships, the expression of *PAX3/7* and *MSX1* was higher than that of *FOXD3*, *ETS1*, and *SNAI2*[50]. Furthermore, the downregulation of *CDH2* coincided with the expression of *SNAI2* and *LMO4*[50]. Changes in cadherin expression in the aggregates were analyzed by IHC. Epithelial cadherin CDH1 was expressed in the aggregates almost entirely on day 2 and then gradually restricted as differentiation proceeded (Fig. 5b). Most CDH1⁻ regions were occupied with FOXD3⁺ cells. SOX10⁺ cells in day-4 aggregates expressed the EMT-related cadherin CDH6 with a complementary pattern to CDH1, indicating the switching of cadherins during NCC-like cell differentiation (Fig. 5c). These results suggest that the in vitro induction of NCC-like cells may recapitulate the in vivo developmental processes to some extent (Fig. 5d).

## Induction of branchial arch mesenchyme-like aggregates with specific regional identity

We tested the potential of these aggregates as in vitro models of early craniofacial development. As earlier mentioned, the expression of *DMBX1*, *LHX5*, *OTX2* (Supplementary Fig. 8b), and ETS1 (Fig. 5a) suggested that the induced NCC-like cells possessed a cranial identity (Fig. 5a, Supplementary Fig. 8b)[35,36]. cNCCs migrating into future craniofacial regions are *HOX*-negative in vivo[51]. We were unable to detect *HOX*-positive NCC-like cells in the aggregates, indicating that the cells in the aggregates possessed the ability to differentiate into the craniofacial mesenchyme (Supplementary Fig. 12).

We performed a long-term culture of the aggregates to induce craniofacial mesenchyme-like cells. The addition of basic fibroblast growth factor (bFGF) from day 4 enabled long-term culture, preventing the collapse of the aggregates during the culture period (Fig. 6a). bFGF-treated aggregates expressed the craniofacial mesenchyme markers *PRRX1* and *TWIST1* at higher levels than untreated aggregates (Fig. 6b)[52]. IHC of day-28 aggregates revealed that many cells expressed these genes, indicating the induction of craniofacial mesenchymal-like cells (Fig. 6d, e). The branchial arch markers, DLX2 and *BARX1*, were also expressed in day-28 aggregates (Fig. 6b, e, f)[53]. These aggregates also expressed LHX6, a specific marker expressed on the oral side of the first branchial arch mesenchyme in a spatially restricted manner, as observed in vivo (Fig. 6b, e)[54]. The first branchial arch is composed of the MX and MN, which give rise to the upper and lower jaws, respectively. bFGF-treated aggregates expressed *POU3F3*, which was strongly expressed in early MX[9], but did not express the MN marker *DLX5* (Fig. 6b and Supplementary Fig. 13), indicating that bFGF-treated aggregates acquired an MX-like identity (Fig. 6c).

The choice of fate between MX and MN is regulated by the peptide hormone EDN1[10,55]. EDN1 is secreted from the epithelium surrounding the MN and is essential for the induction of MN specifiers, such as *DLX5/6* (Fig. 7a). Loss of the EDN1 signal results in the transformation of the lower jaw into an upper jaw-like structure[10], whereas overactivation transforms the MX into an MN-like identity[55]. However, the NCC-like cells in the aggregates expressed the EDN1 receptor *EDNRA* (Fig. 7b), and the aggregates comprised only a small number of NNE-like cells, a possible source of EDN1. We hypothesized that low levels of EDN1 in the aggregates would result in the acquisition of an MX-like identity

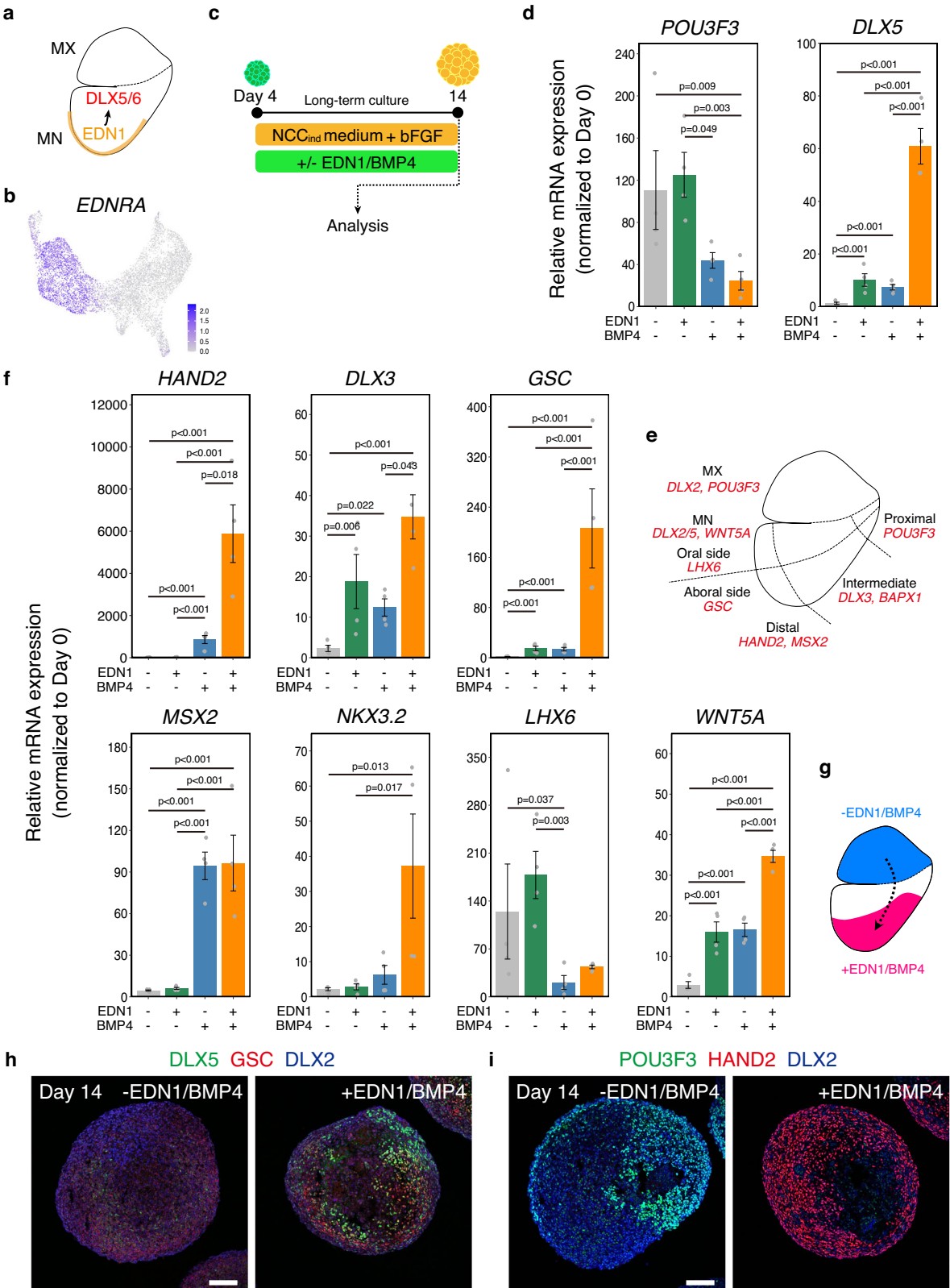

during long-term culture. Therefore, we cultured the aggregates with or without EDN1 and BMP4 from day 4 in the presence of bFGF (Fig. 7c). BMP4 is also secreted from the craniofacial epithelium and is important for the patterning of MN[11]. When EDN1 and BMP4 were added, *POU3F3* was downregulated, whereas *DLX5* was upregulated (Fig. 7d). *DLX5* expression was higher in the aggregates treated with both EDN1 and BMP4 than in those

treated with either EDN1 or BMP4 alone. However, the cooperative effect of EDN1 and BMP4 on *DLX5* expression has not been reported previously.

The MN is divided into several subdomains (Fig. 7e). We analyzed the expression of domain-specific genes to characterize the MN-like cells induced by EDN1/BMP4 (Fig. 7f). EDN1/BMP4-treated aggregates expressed the distal region markers *HAND2*

**Fig. 7 | Mandibular arch (MN)-like differentiation of the aggregates. a** Schema of the expression of EDN1 (endothelin 1) and DLX5/6 in first branchial arch. MX: maxillary arch. **b** Feature plot of EDNRA in scRNA-seq data. **c** Scheme of long-term culture of aggregates. Aggregates were cultured with or without EDN1 and BMP4 (bone morphogenetic protein 4) in the presence of bFGF (basic fibroblast growth factor) from day 4 and analyzed by real-time PCR and immunohistochemistry at day 14. NCC: neural crest cell. **d** Real-time PCR analysis of *POU3F3*, and *DLX5*, which are expressed in MX and MN, respectively. Data are presented as mean ± SEM (*n* = 4 independent experiments). Two-way ANOVA with Tukey–Kramer test was used. Source data are provided as a Source Data file. **e** Schema of the spatial organization of domains in the first branchial arch. **f** Real-time PCR analysis of the region-specific

genes. Data are presented as mean ± SEM (*n* = 4 independent experiments). Two-way ANOVA with Tukey–Kramer test was used. Source data are provided as a Source Data file. **g** Hypothetical representation of the effect of EDN1 and BMP4 on regional identity of the aggregates. The aggregates switched their identity from the MX-like to the MN-like, when treated with EDN1 and BMP4 during the long-term culture. **h** Immunostaining of non-treated aggregates and EDN1/BMP4-treated aggregates on day 14 with antibodies for DLX5 (green), GSC (red), and DLX2 (blue). Scale bar, 100 μm. **i** Immunostaining of non-treated aggregates and EDN1/BMP4-treated aggregates on day 14 with antibodies for POU3F3 (green), HAND2 (red), and DLX2 (blue). Scale bar, 100 μm. Three independent cultures were used for experiment and representative images are shown.

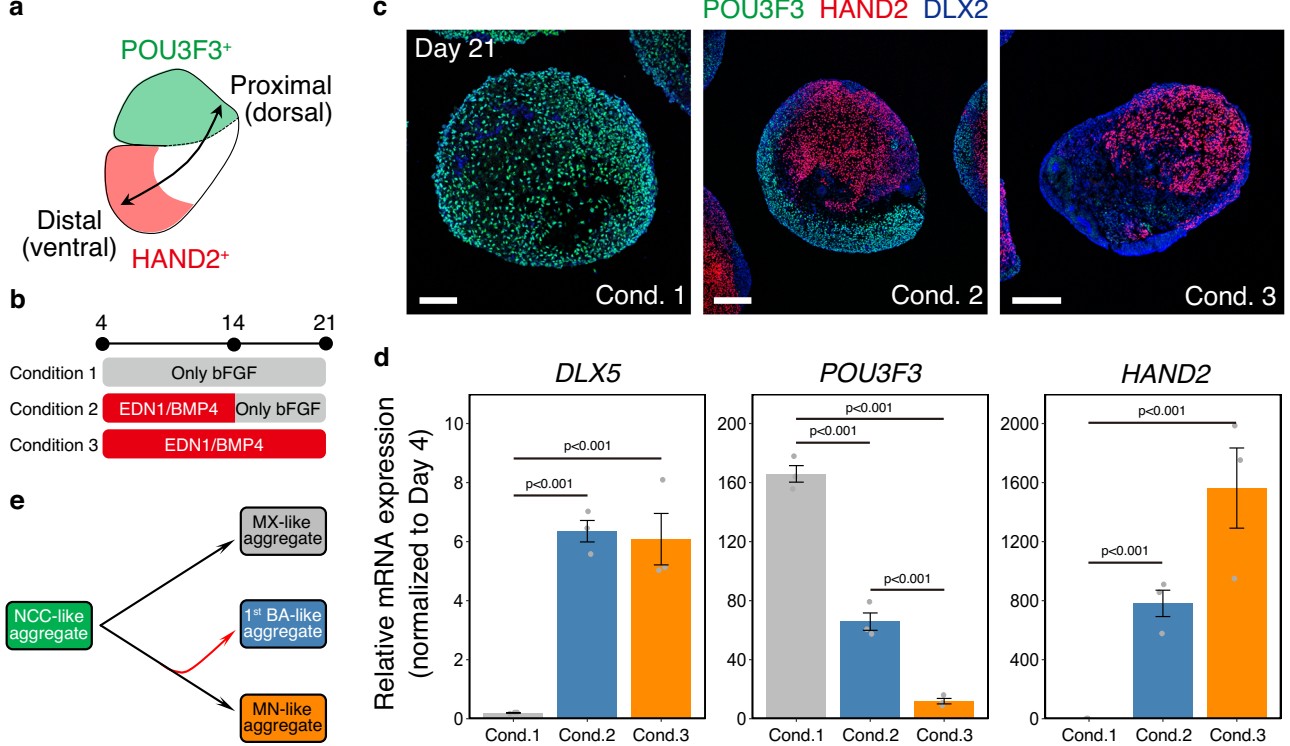

**Fig. 8 | Formation of the 1st BA (first branchial arch)-like aggregates containing both proximal and distal-type cells. a** Schema of the axial organization of the 1st BA. The proximal (dorsal) side is *POU3F3*⁺ MX (maxillary arch), and the distal (ventral) side is *HAND2*⁺ distal MN (mandibular arch). **b** Schematic of experiment. The duration of treatment with EDN1 (endothelin 1) and BMP4 (bone morphogenetic protein 4) differed among conditions 1, 2, and 3. The aggregates were analyzed using real-time PCR and immunohistochemistry on day 21. bFGF: basic fibroblast growth factor. **c** Immunostaining of the day-21 aggregates cultured in each condition with antibodies for POU3F3 (green), HAND2 (red), and DLX2 (blue). Scale bars, 100 μm (the left one) and 200 μm (the others). Three independent

cultures were used for experiment and representative images are shown. Cond. 1–3 condition 1–3. **d** Real-time PCR analysis of the region-specific genes. Data are presented as mean ± SEM (*n* = 3 independent experiments). One-way ANOVA with Tukey–Kramer test was used. Source data are provided as a Source Data file. **e** Scheme for differentiation of first branchial arch-like aggregates. When cultured with bFGF alone, the aggregates differentiated into MX-like states, whereas they differentiated into MN-like states in the presence of EDN1 and BMP4. Temporary treatment with EDN1 and BMP4 induced the formation of single aggregates containing both *POU3F3*⁺ Mx-like and *HAND2*⁺ Mn-like regions.

and *MSX2*. However, *MSX2* expression was upregulated when BMP4 was added alone. EDN1/BMP4-treated aggregates also expressed the intermediate region markers *DLX3* and *NKX3.2*[56]. Regarding the oral-aboral axis, the aboral marker *GSC* was upregulated in the EDN1/BMP4-treated aggregates, whereas the oral side marker *LHX6* was downregulated. Furthermore, in the EDN1/BMP4-treated aggregates, *WNT5A*, which regulates the proper morphogenesis of MN, was also upregulated[57]. We confirmed the expression of these markers in EDN1/BMP4-treated aggregates using IHC (Fig. 7h, i). These results suggested that the aggregates were similar to the in vivo branchial arch mesenchyme in their responsiveness to signaling factors, and that EDN1 and BMP4 cooperatively induced the identity of the aboral side of the distal, intermediate regions of the MN (Fig. 7g).

## Formation of the patterned branchial arch-like aggregates through temporary treatment

The MX and MN can be recognized as the dorsal and ventral sides of the same tissue, as the most dorsal side is *POU3F3*⁺ MX and the ventral side is *HAND2*⁺ region of the MN (Fig. 8a). At day14, in bFGF-treated aggregates that differentiated into an MX-like state, most NCC cells still expressed SOX10 (Supplementary Fig. 14). In contrast, in EDN1/BMP4-treated aggregates that differentiated into an MN-like state, most mesenchymal cells expressed DLX5 instead of SOX10, indicating fate commitment into the MN-like lineage. However, in these aggregates, a subset of cells still expressed SOX10, suggesting that they were not committed to the MN-like lineage and could differentiate into the MX-like state if EDN1 and BMP4 were removed from the medium. We tested

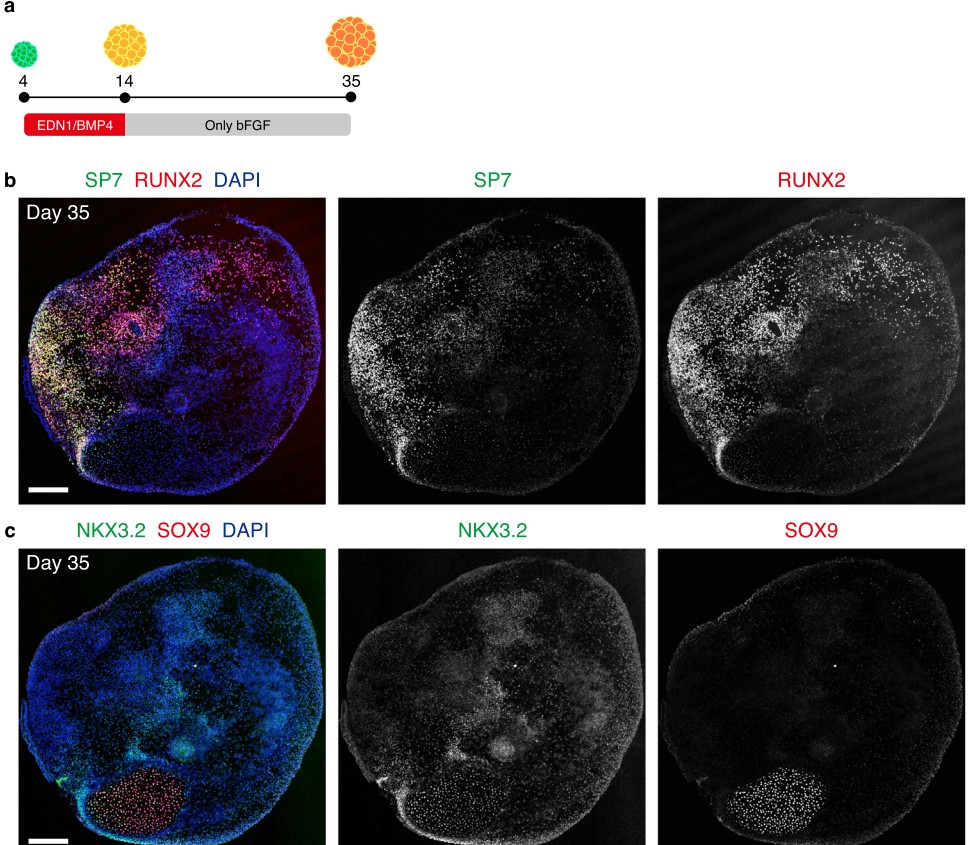

**Fig. 9 | Osteogenic/Chondrogenic differentiation in the aggregates after long-term culture. a** Schema of the long-term culture. After the wash out of EDN1 (endothelin 1) and BMP4 (bone morphogenetic protein 4), the aggregates were cultured in NCC$_{ind}$ medium supplemented with only bFGF (basic fibroblast growth factor). NCC neural crest cell. **b** Immunostaining of the day-35 aggregates with antibodies for SP7 (green), RUNX2 (red), and counterstaining with DAPI (blue).

Scale bar, 200 μm. Three independent cultures were used for experiment and representative images are shown. **c** Immunostaining of the day-35 aggregates with antibodies for NKX3.2 (green), SOX9 (red), and counterstaining with DAPI (blue). Scale bar, 200 μm. Three independent cultures were used for experiment and representative images are shown.

whether single aggregates contained both POU3F3$^+$ (dorsal MX-like) and HAND2$^+$ (ventral MN-like) cells by removing EDN1 and BMP4 from the medium on day14 (Fig. 8b). The temporarily treated aggregates (condition 2 in Fig. 8b) expressed both *POU3F3* and *HAND2* at higher levels than the EDN1/BMP4-treated (condition 3 in Fig. 8b) and non-treated aggregates (condition 1 in Fig. 8b) on day 21(Fig. 8c). Furthermore, POU3F3$^+$ and HAND2$^+$ cells formed spatially separated domains in temporarily treated aggregates, indicating mutually exclusive fate choice and self-patterning properties of cells (Fig. 8d).

We analyzed previously reported scRNA-seq data from the mouse branchial arch on embryonic day 9.5 (E9.5) as a reference[58]. *Hox*-negative cells were extracted from NCC clusters and re-clustered (Supplementary Fig. 15a). There were clusters positive for the MN markers *Dlx5* and *Hand2* (clusters 1, 2, and 7) and clusters positive for the MX arch marker *Pou3f3* (clusters 0, 3, 5, and 6). Several presumptive markers for these clusters were identified (Supplementary Fig. 15b). On day 21, EDN1/BMP4-untreated aggregates expressed the presumptive MX arch markers *EBF1* and *POU3F1* at higher levels, and EDN1/BMP4-treated aggregates expressed the presumptive MN marker *DLK1* at higher levels (Supplementary Fig. 15c). Temporarily treated aggregates expressed high levels of both markers. The scRNA-seq of E9.5 mouse embryo showed that *Pou3f3* and *Hand2*$^+$ were expressed in a mutually exclusive manner (Supplementary Fig. 15d). IHC also showed the separation of Pou3f3$^+$ and Hand2$^+$ cells in the first branchial arch at E9.5, as in case of temporarily treated aggregates (Supplementary Fig. 15e).

As the branchial arch mesenchyme differentiates into craniofacial bones and cartilage in vivo[1], we analyzed the osteogenic/chondrogenic potential of the aggregates by extending the culture period (Fig. 9a). On day 35, temporarily treated aggregates expressed RUNX2, a marker for osteogenic/chondrogenic lineage cells (Fig. 9b)[59]. Some RUNX2$^+$ cells express SP7, an osteogenic cell marker, indicating their differentiation into this lineage[59]. Furthermore, in the aggregates, there were cell condensates positive for SOX9 and NKX3.2, which were expressed in the Meckel's cartilage of mouse embryos (Fig. 9c)[60,61]. These results suggest spontaneous differentiation of NCC-like cells into osteogenic/chondrogenic lineages in culture.

## Discussion

In this study, we established a method for inducing aggregates enriched with NCC-like cells to differentiate into a brachial arch-like state. In this method, NCC-like cells can be induced through a simple procedure within 5 days with efficiencies of 45.4–76.5% depending on the cell lines (Supplementary Fig. 3c). Aggregate-intrinsic WNT/BMP signaling can induce NCC-like cells in an NPB-like state. NPB-like cells also differentiate into other ectodermal lineages, including NEC-like-and NNE/PPE-like cells. This feature may be suitable for investigating the mechanisms underlying the fate choice in multipotent NPB-like cells in a natural context.

Although the expression of several WNTs and BMPs was upregulated from differentiation day 3 in the aggregates, the inhibition of these signals from day 3 did not affect the induction of NCC-like cells. We could not determine the cause of this

ineffectiveness of Wnt/BMP inhibition from day 3; however, the autoregulation of NCC specifiers could possibly facilitate the differentiation of NCC-like cells in the absence of Wnt/BMP signals after day 3. For example, *Pax3* specifies NCC identity in *Xenopus* animal caps and maintains its expression through autoregulatory feedback[62]. Based on our scRNA-seq data, only 27.7% of NPB-like cells in day-2 aggregates expressed *PAX3*, and 86.5% in the day-3 aggregates (Supplementary Fig. 16a). Although *Pax3* required *Zic1* to activate the complete NCC developmental program in *Xenopus*[62], most NPB-like cells in the aggregates expressed the *ZIC2/3/5* family genes (Supplementary Fig. 16b). *Zic2/3/5* are also important for the development of NCCs in model animals[63–65]. Considering these factors, it is possible that NCC specifiers, such as *PAX3* and *ZIC2/3/5*, which were already expressed by day 3, could promote the differentiation of NPC-like cells into NCC-like cells independently of WNT/BMP signals. As the WNT signal regulates multipotency of NCCs through the *Lin28/let-7* axis in chick[66], it is possible that NCC-like cells in the aggregates treated with IWP-2 from day 3 have deficits in terms of differentiation potential, although we have not checked that possibility yet. As for BMP signals, reporter assays using BMP-responsive elements have revealed that the activity of the BMP signal was relatively low during the specification of NCCs and was upregulated during EMT in chick[67]. This was consistent with the ineffectiveness of BMP inhibition from day 3 on NCC-like cell induction in the aggregates.

We also analyzed the effect of activin/TGFß inhibition by SB in our protocol. NCC induction methods based on dual-SMAD inhibition use SB as an important component, but SB showed adverse effects on NCC induction, as reported by Leung and colleagues[5]. In our protocol, SB had a positive impact on NCC induction, as indicated by the enhanced generation of FOXD3+ cells and the inhibition of SOX17+ endoderm induction on day 4. It is important to elucidate the cause of the difference in the response to SB to understand the mechanisms behind the induction of NCC from hPSCs.

scRNA-seq analysis suggested a possible lineage bifurcation from the NPB-like state to the NCC-like, NEC-like, and NNE/PPE-like states in the aggregate, which is consistent with the fact that NPB cells can contribute to all ectodermal lineages in model animals[32–34]. We identified genes that were upregulated during the differentiation into specific lineages. However, the functions of some of these genes in NPB-like cells have not yet been elucidated. Further analysis of these uncharacterized genes would provide a more precise understanding of the lineage specifications in multipotent NPB-like cells.

Long-term culture of the aggregates demonstrated their potential to differentiate into an MX-like state in the presence of bFGF. The aggregates tended to collapse during long-term culture without bFGF, consistent with the previous observation that mesenchymal cells in the first branchial arch require the FGF signal to survive[54].

MX is transformed into an MN-like state upon receiving an EDN1 signal in vivo[55]. The aggregates express marker genes for MN, such as *DLX5* and *HAND2*, when stimulated with EDN1 and BMP4. The cooperative effect of EDN1 and BMP4 on *DLX5* expression has not been observed in previous in vivo studies, indicating an advantage of this in vitro model over in vivo models.

Temporary treatment with EDN1 and BMP4 induced the formation of both HAND2+ and POU3F3+ cells within single aggregates. Although the mechanism underlying the co-emergence of HAND2+ cells and POU3F3+ cells has not yet been identified, it is possible that fate-uncommitted NCC-like cells in day-14 EDN1/BMP4-treated aggregates differentiated into an MX-like state after removing EDN1 and BMP4 from the medium (Supplementary Fig. 14). Although EDN1 and BMP4 were both applied to the medium, HAND2+ and POU3F3+ cells did not intermingle but segregated spatially, forming distinct domains inside the aggregates. The mechanisms underlying the segregation of the two

cell types have not yet been elucidated. The gathering of the same type of cells through a chemoattractant and dominant adhesion between these cells might be considered possible mechanisms. For example, the NCC-like cells expressed *WNT5A* after stimulation with EDN1/BMP4 (Fig. 7f). *WNT5A* acts as a chemoattractant[68]. If MN-type cells respond selectively to WNT5A, the chemotactic properties of WNT5A may cause MN-type cells to form clusters in aggregates. In the EDN1/BMP4-treated aggregates, DLX5+ cells expressed CDH11 strongly on day 7 (Supplementary Fig. 17). *CDH11* expression appears to be upregulated in the MN, MX, and limbs at the onset of their morphogenesis[69]. CDH11 exhibits homophilic binding when expressed in L cells, and CDH11+ cells are sorted from the negative population in aggregates derived from the dissociated limb bud mesenchyme[69]. Other factors beyond *CDH11* may be at play; moreover, such a homophilic cell adhesion-dependent mechanism might also be involved in the formation of HAND2+ and POU3F3+ regions in the aggregates.

Finally, differentiation into the osteogenic/chondrogenic lineages was observed in the aggregates, expanding the applicability of this model. Although the important genes and signals for commitment to these lineages have been identified[70], the actual processes during cell fate decisions are not yet properly understood. Using appropriate reporter cell lines, the dynamics of commitment to the osteogenic or chondrogenic lineage can be visualized. Such experiments could also provide insights into the cellular mechanisms underlying craniofacial development.

## Methods

### Maintenance and differentiation of hPSCs

StemFit AK02N (Reprocell) was used as a maintenance medium for hPSCs in feeder-free cultures. Three PSC lines were used in the experiments. Two ESC lines (KhES-1 and KthES11) and one iPSC line (253G1). For passage, the cells were dissociated into single cells by treatment with 0.5 × TrypLE Select (Gibco) diluted with 0.5 mM EDTA/phosphate-buffered saline (PBS) ($-Ca^{2+}$, $-Mg^{2+}$). Dissociated cells were counted and $1.6 \times 10^4$ cells were plated into the wells of 6-well culture plates coated with iMatrix-511 (Nippi). Y-27632 was added (10 μM) on the first day of plating. For differentiation, the dissociated cells were suspended in NCC$_{ind}$ medium containing Y-27632 ($1.0 \times 10^4$ cells per 100 μl per well) and re-aggregated in 96-well V-bottom low-cell-adhesion plates (Sumitomo Bakelite). This day was defined as day 0. Briefly, 50 μl NCC$_{ind}$ medium not containing Y-27632 was added to each well on day 1 and half of the medium was replaced with fresh medium on day 3.

For long-term culture, 50 μl of NCC$_{ind}$ medium containing bFGF was added on day 4. Subsequently, half of the medium was replaced with the bFGF-containing NCC$_{ind}$ medium every other day. In some experiments, EDN1 and BMP4 were added from day 4 onward. For temporary treatment with EDN1 and BMP4, the aggregates were washed with fresh NCC$_{ind}$ medium and cultured in bFGF-containing NCC$_{ind}$ medium.

The NCC$_{ind}$ medium was DMEM/F-12 with GlutaMAX (Gibco) supplemented with 1 × B-27 minus vitamin A (Gibco) and 5 mg/ml bovine serum albumin (BSA; Fujifilm). KSR (Invitrogen) was added as required.

For neuronal, glial, and melanocytic differentiation, cells dissociated from Day-5 aggregates were plated on iMatrix-coated glass-bottom dishes ($1.0 \times 10^4$ cells/cm²) and cultured in NCC$_{ind}$ medium. Aggregates were dissociated using TrypLE Express (Gibco). For differentiation into smooth muscle cells, the cells were plated on glass-bottomed dishes coated with Geltrex (A1413202; Gibco). For differentiation into osteogenic, chondrogenic, and adipogenic cells, dissociated cells were cultured on Geltrex-coated dishes ($1.3 \times 10^4$ cells/cm²) in NCC$_{ind}$ medium containing bFGF for 6 days. The cells were dissociated using Accumax (Nacalai Tesque) and used in subsequent cultures. For

osteogenic differentiation, the cells were plated on Geltrex-coated dishes ($1.0 \times 10^4$ cells/cm$^2$) and cultured with the contents of the StemPro Osteogenesis Differentiation Kit (Gibco). For chondrogenic differentiation, the dissociated cells were aggregated in 96-well V-bottom low-cell-adhesion plates ($3 \times 10^5$ cells/well) and cultured using the contents of the StemPro Chondrogenesis Differentiation Kit (Gibco). For adipogenic differentiation, the cells were plated on Geltrex-coated dishes ($1.0 \times 10^4$ cells/cm$^2$) and cultured with the contents of the StemPro Adipogenesis Differentiation Kit (Gibco).

Bright-field images were acquired using a BZ-X710 microscope with the BZ-X Viewer (Version 1.4.0.1, KEYENCE).

The concentrations of reagents in the culture medium during the experiment were as follows: Y-27632 (10 μM; Tocris), dorsomorphin (2 μM; R&D Systems), IWP-2 (5 μM; Selleck Biotech), SB-431542 (10 μM; Selleck Biotech), bFGF (20 ng/ml; R&D Systems), EDN1 (25 ng/ml; Peptide Institute, Inc.), and BMP4 (5 ng/ml; R&D Systems).

The experimental protocols were approved by the Research Ethical Committee of Kyoto University.

## Immunohistochemistry and other staining

For sectioning of the aggregates, the aggregates were fixed with 4% paraformaldehyde/PBS (-) at 4 °C for 15 min, washed with PBS, and immersed in 15% sucrose/PBS (-) at 4 °C overnight. The fixed aggregates were frozen in O.C.T. compound (Sakura Finetek) and sectioned using a cryostat (10 μm sections). The sections were permeabilized with 0.3% Triton X-100/PBS (-) for 20 min at room temperature and then blocked with Blocking-One (Nacalai Tesque) for 30 min at room temperature. The primary antibodies were diluted in Blocking-One and incubated overnight at 4 °C with the sections. After washing, the sections were incubated with diluted secondary antibodies (Donkey-raised antibodies conjugated with Alexa Fluor 488 or Cyanine Cy3 or Cyanine Cy5, Jackson ImmunoResearch Labs) and 4′,6-diamidino-2-phenylindole (DAPI; Nacalai Tesque) for 2 h at room temperature. A TCS SP8 confocal microscope and the Leica Application Suite X software (Version 3.5.5.19976, Leica) was used to capture the images. ImageJ (1.53k) was used to merge single-channel images. The dissociated cells cultured in glass-bottomed dishes were stained in a similar manner. Cryoprotection and sectioning steps were omitted. For preparing mouse embryonic tissue, fixation period was prolonged to 2 h, and cryoprotection was performed with 30% sucrose/PBS (-). The antibodies used in this study are listed in Tables S1 and S2.

For Alizarin Red staining and Oil Red O staining, the cells were fixed in 4% paraformaldehyde/PBS (-) for 15 min at room temperature. For Alcian Blue staining, the aggregates were fixed in 4% paraformaldehyde/PBS (-) at 4 °C for 15 min and cryoprotection and sectioning were performed in the same manners with IHC samples.

All animal procedures comply with all relevant ethical regulations and guideline for animal studies approved by the Research Ethical Committee of Kyoto University. Pregnant mice (Slc:ICR) were obtained from SHIMIZU Laboratory Supplies Co., Ltd. The mice were maintained on a 12-h light/dark cycle at room temperature ($24 \pm 2$ °C) with constant humidity ($55 \pm 10\%$).

## Flow cytometry analysis

The aggregates were pooled and dissociated using TrypLE Express (Gibco) at 37 °C for 8 min. The reaction was stopped by adding PBS, and the cells were filtered through a cell strainer. Dissociated cells were stained with an allophycocyanin-conjugated anti-p75 antibody or isotype control (1:50; Miltenyi Biotec) according to the manufacturer's instructions. After washing, the cells were counted using a FACSAria IIIu flow cytometer (Becton Dickinson). The data were analyzed using the FACSDiva software (Version 6.1.3, Becton Dickinson). The FlowJo software (Version 10.8.1, Becton Dickinson) was used to make histograms. The antibodies used in this study are listed in Table S3.

## RNA isolation and real-time PCR

Total RNA was purified from 24 cell aggregates per condition using the RNeasy Micro Kit (QIAGEN), and cDNA was synthesized using SuperScript II (Invitrogen). Real-time PCR was performed using Power SYBR Green PCR Master Mix and QuantStudio 6 and 7 Flex Real-Time PCR Systems (Version 1.3, Applied Biosystems). All data were analyzed using the standard curve method and *GAPDH* expression was used for normalization. The primer sequences are summarized in Table S3.

## scRNA-seq and data analysis

Aggregates derived from KhES-1 were used for scRNA-seq experiments. For sample preparation, 32 aggregates were pooled and dissociated using TrypLE Express for 8 min at 37 °C. Gentle pipetting with a low-binding tip was performed 4 and 8 min after incubation. An equal amount of 0.04% BSA/PBS (-) was added to the dissociated cells, and the cell suspension was filtered through a cell strainer (Falcon). The cells were counted and pelleted through centrifugation at $430 \times g$ for 5 min. The cells were resuspended in 0.04% BSA/PBS (-) (adjusted to 1000 cells/μl) and placed on ice until use. About 5000 cells per sample were used in library construction using v3 Chromium Single Cell 3′ Reagent Kits (10X Genomics). Gel bead-in-emulsion generation, cDNA synthesis, and library preparation were performed according to the manufacturer's instructions. The constructed libraries were sequenced on the NovaSeq 6000 platform (Illumina).

Cell Ranger 5.0.1 was used to process the raw data from the NovaSeq 6000. Next, we analyzed the processed data using Seurat (Version 4.3.0.1) on R (Version 4.3.1). Briefly, the cells with aberrantly high or low numbers of unique molecular identifiers (UMIs) and high mitochondrial gene detection rates were removed. Gene expression levels were normalized to the total number of UMIs in each cell line. The cell cycle score was calculated using the expression levels of cell cycle-related genes to perform data scaling using cell cycle regression. The detailed process followed the developer's instructions (https://satijalab.org/seurat/archive/v3.1/cell_cycle_vignette). Subsequently, a principal component analysis and unsupervised clustering were performed. To merge the scRNA-seq data of samples from different days, we applied the "merge" function. monocle3 (Version 1.3.3) was used for the pseudotime analysis. Pseudotime values were assigned to each cell by "learn_graph" and "order_cells" functions. Genes expressed differentially along inferred pseudotime were identified by "graph_test" function. For RNA velocity analysis, loom files were generated using Velocyto (Version 0.6). RNA velocity analysis was conducted using scVelo (Version 0.2.4) on Python (Version 3.9.5). RNA velocity was estimated by using the dynamic model. The candidates of lineage-specific driver genes were identified by "scvelo.tl.rank_velocity_genes" module. For trajectory analysis, a subset of the merged samples was processed using slingshot (Version 2.8.0) and tradeSeq (Version 1.1.4) on R. R packages dplyr (Version 1.1.3), ggplot2 (Version 3.4.3) scCustomize (Version 1.1.3) stats (Version 4.3.1) tidyr (Version 1.3.0) and Python packages anndata (Version 0.7.8), numpy (Version 1.20.3) pandas (Version 1.4.1), scanpy (Version 1.8.2) scipy (Version 1.8.0) were also used for data processing and visualization.

## Statistics

Data are presented as the mean ± standard error of the mean (SEM). Significant differences were determined using two-tailed *t*-tests or ANOVA tests followed by Tukey–Kramer test. Microsoft Excel for Mac (Version 16.44) was used for data preprocessing and formatting.

## Reporting summary

Further information on research design is available in the Nature Portfolio Reporting Summary linked to this article.

## Data availability

The scRNA-seq data generated in this study have been deposited in the Gene Expression Omnibus (GEO) under accession code GSE199158. Source data are provided with this paper.

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

## Acknowledgements

We thank all members of the Eiraku Lab for their productive discussions. We are especially grateful to Fumi Wagai for her generous assistance in continuing the laboratory activities. We also thank the Single-cell Genome Information Analysis Core (SignAC) at WPI-ASHBi, Kyoto University, for their support with NGS experiments. This work was supported by Grant-in-Aid for Scientific Research on Innovative Areas (Ministry of Education, Culture, Sports, Science, and Technology, MEXT), Japan (16H06480 to M.E.), Grant-in-Aid for Transformative Research Areas (Ministry of Education, Culture, Sports, Science, and Technology, MEXT), Japan (23H04933 to M.E.), and Core Research for Evolutional Science and Technology (CREST, JST) (JPMJCR12W2 to M.E.).

## Author contributions

Y.S. and M.E. conceptualized the study. Y.S. designed and conducted the experiments and performed data analysis. R.O. and K.T. performed the experiments. Y.S. wrote the manuscript. All coauthors contributed to the respective text parts, data analyses, and revised the manuscript.

## Competing interests

The authors declare no competing interests.
