## [Peer Review File · Nature Communications]

In vitro induction of patterned branchial arch-like aggregate from human pluripotent stem cellsREVIEWER COMMENTS

Reviewer #1 (Remarks to the Author):

The Article “In vitro induction of patterned branchial arch-like aggregate from human pluripotent stem cells” by Yusuke Seto and Colleagues is very interesting and is likely to be useful for the NC community at large. They present sufficient evidence to suggest that cranial NCC can be induced in cultures of aggregated PSC using media devoid of exogenous WNT and BMP signal. The induction occurs in an efficient manner using 3 cell lines (2 hES and 1 iPSC) in just 5 days. The cultures do not render homogeneous population, with 100% of the cells undergoing NCC induction, however the numbers are robust enough. Furthermore the authors show that these cells acquire a cranial character belonging to the 1st branchial arch and that they can be directed to differentiate into mandibular, maxilar or both enabling multiple future explorations of the mechanisms at play. The figures are clear and provide strong evidence of the claims.

1st There are just a few suggestions and comments. The manuscript contains lots of typos and grammatical errors.

2nd While at some point a reference is made to model organisms, but is missing in other places, for example, BMP and WNT have been shown to contribute to NC development in multiple organisms, statements along with adequate references are necessary.

3rd Interesting that WNT and BMP upregulation is shown at or after day 4, but according to the temporal inhibition, they trigger effects at earlier stages...this needs to be discussed, as irt does not match with the commitment suggested by authors.

4th Today multiple human NC models also include TGFb inhibition, which here is not used but should be discussed

The following notes direct to minor suggestions:

Page 4 line 93: The expression of both FoxD3 and Sox10 appears close to each other in model organisms and human embryos. Earlier NC markers include Pax3, Pax7, Sox9, etc.

Page 5 Line 97...”neural cells express p75 at high and low levels, respectively 3”. Betters et al (Developmental Biology 344 (2010) 578–592) showed in 2010 the lack of NC specificity of P75 for neurla crest cells in human embryos, and thus this seem appropriate to include this earlier reference.

Lines 98-99: Supp Fig1 D does not show P75 analyses.

Line 113-114: ... “confirmed the presence of those types of cells in the aggregates by immunohistochemistry (Fig. 1b, 2e, f).” No immunos corresponding to this panels or stainings found. Only Six1 and Isl 1/2.

Lines 115-116: “Unlike other NCC induction methods from hPSC, extrinsic manipulation of WNT/BMP signals was not necessary in our method 5. Include Hackland et al 2019 (Stem Cell Reports j Vol. 12 j

920–933) whci modulates BMP, and WNT. Leung et all, relies on WNT only.

Page 9, lines 206-207: “POU3F3 which is strongly expressed in the maxillary arch 34, but they did not express DLX5 which is expressed in the mandibular arch (Fig. 6c).” Adding the fluorescence from the supplemental figure showing the negative expression by Immuno is relevant, particularly given that the levels of the Q-RT-PCR are so close.

Page 12 Lines 206-207: “POU3F3 which is strongly expressed in the maxillary arch 34, but they did not express DLX5 which is expressed in the mandibular arch (Fig. 6c).” This tentative sentence should be moved to the discussion and should be followed by a more informative speculation.

Reviewer #2 (Remarks to the Author):

Seto et al reports a novel in human vitro model that generates 3-dimensional aggregates containing neural crest cells (NCCs) from pluripotent stem cells. Temporal scRNAseq analysis revealed that NCCs were specified from an intermediate neural plate border (NPB) stage followed by an EMT event. Further, prolonged cultivation under exposure to EDN1/BMP4 treatments resulted in acquisition of pharyngeal arch (PA) identity, and both mandibular and maxillary arches (derived from the first PA) within the aggregates. These observations suggest recapitulation of key developmental events, providing opportunities for the investigation of human NCC development, including cell patterning and acquisition of positional identity. In my opinion, this is relevant for 3 reasons: 1) early specification and patterning events during craniofacial development are complex processes that still need clarification; 2) the strategy could be used not only to bring insights into early human NCC development (for which models are scarce), but also into the etiology of several congenital disorders, which often present with mandibular and maxillary defects; 3) the simplicity (and probable cost effectiveness) of the model makes it quite accessible.

This work has great potential to contribute to the field, but it is lacking in some aspects in its current form. Such contributions and impact seem a bit underappreciated throughout the text, and perhaps a more solid demonstration of its potential to bring new insights could significantly improve the manuscript. Please find below some issues I think should be resolved:

1) General writing – authors should consider revising the manuscript to resolve grammatical/stylistic issues. Some sections are not clear, for example:

- a) Abstract, line 36 - I presume you mean “mean specification and patterning events”
- b) Introduction, lines 39-41 – This section sounds odd, especially the terms used (e.g. facial identifiers)
- c) Introduction, line 44 - “The NCCs are transient cell population which arises from neural plate border (NPB). They migrate from dorsal their origin to ventral craniofacial primordium”. Did you mean “They migrate from their dorsal origin to ventral craniofacial primordia”?
- d) Introduction, line 62 – “...their capability to directionally differentiate into NCC”. Do authors mean “directed differentiation” or “direct differentiation”?

2) Introduction, lines 52-56 - This section seems oversimplified. Reference 2 reports that postmigratory NCC movements within the PA are involved in a dynamic cell fate specification process, which takes place via sequential fate bifurcations as opposed to specification in discrete compartments of the PA, as previously thought. Further, these observations were made in postmigratory NCCs, so stating that 'migration of NCCs is involved in that process' is misleading as it may refer to migratory NCCs.

3) Results, lines 97-99 - "On day 5, approximately 70% of cells in the aggregates were p75^{bright}, although the percentage varied among cell lines (Fig. 1d and Suppl. Fig. 1D)." Variation in p75 dim/bright in flow cytometry can be indicative of variation in the proportion of NCC vs neural cells, but the immunostaining of cell aggregates (Suppl. Fig. 1D) show CDH1⁺ clusters (neither a neural or NCC marker). Please make a distinction between these two lines of evidence to support your conclusion here.

4) Please specify which cell line(s) was used in scRNAseq assays. Did you assess p75 expression in your scRNAseq data (Fig. 2b)? This could indicate whether flow cytometry for p75 alone has sufficient resolution to estimate the % of NCCs. p75 is a membrane receptor, so this could be informative to researchers attempting to sort NCCs using this experimental framework.

5) Line 106, "were classified as SOX10⁺ NCCs (Fig. 2b)". Fig. 2c must be referenced here as well.

6) Fig 3b: The layout makes it difficult to read panel b.

7) Lines 122-124, referring solely to SOX10 expression upon WNTi/BMPi treatment: This is also valid for PAX6, SIX1 and EYA1.

8) Lines 124 – 127: Please specify to which treatment day you are referring (for CDH1 EYA1 and PAX6)

9) Fig 3d: Please provide magnifications

10) Lines 131-133 "We found that, a lot of cells in SIX1⁺ PPE ...expressed PAX6" (Fig. 3e): Don't you mean SIX1⁺/CDH1⁺? CDH1 is expressed in the PPE and magnifications in Fig. 3e show CDH1⁺ regions. Please remove comma after "We found that".

11) Figure 4: In Fig. 4b, IRX1 and SP5 do not seem specific to the NPB (clusters 5 and 10). The NPB expresses a variety of markers of all ectodermal derivatives, so I understand it might be difficult to bring forward specific expression patterns. Still, it is known that the NPB expresses neural and non neural transcription factors with some level of co-localization (e.g. Tfap2a/Pax7; Tfap2a/Sox2, etc). What other NPB markers are expressed in these clusters? Did you observe these co-expression patterns? It would greatly add to the manuscript if authors provided a more in-depth analysis and characterisation of the NPB clusters, as fate specification within the NPB is an active area of research with very little data on human cells.

Check e.g. Williams et al., 2022 and Roellig et al., 2017:

<https://elifesciences.org/articles/21620>

<https://elifesciences.org/articles/74464>

Besides clusters 5,10, 11, what other cell populations were identified here? Can you observe cell fate trajectories from NPB to NCC, PPE and Neural Plate lineages in any specific order? Which expression patterns characterize these trajectories? Can novel putative specification factors be identified under this approach?

12) Fig 4c: what is being indicated by the colours?

13) Line 162: Please specify which GO categories were enriched, and in which cluster(s)/differentiation days. It would be nice if you could show that the time-series analysis of EMT markers replicate some or all of the temporal expression patterns observed in vivo.

14) Fig 7d and others; Multiple comparisons in the manuscript: consider using ANOVA +posthoc tests.

15) Lines 266 – 274: This part sounds a bit vague. Please provide a reference for cadherin-mediated cell sorting and briefly explain how your data may recapitulate this process. Also, how is this related to the higher in vivo expression of CDH11 in the mandibular arch? Please clarify the conceptual basis for this statement.

16) Suppl. Fig. 3: Please include magnifications showing DLX5+/CDH11+ cells. These aggregates were exposed to which bFGF/EDN1/BMP4 treatment regime?

17) Discussion, lines 278-280: The efficiency of NCC generation achieved by the protocol was 45.4-76.5% considering only p75 expression, and assuming p75 alone is sufficient to identify NCCs (which has yet to be demonstrated); and 70.3% according to scRNAseq data. The reported efficiency seems to lie below the latest methods for NCC generation, at least in 2D systems. Please clarify which work is being referenced here.

Moreover, since this is a novel NCC induction strategy, their ability to terminally differentiate into NCC derivatives should be demonstrated.

18) Lines 282-283: Please, elaborate on why the reported in vitro model is more 'beneficial' for investigating early NCC specification. What are the advantages over current methods? How does it compare to current in vitro/in vivo models?

19) Lines 300-303: It is not completely clear how the method of BMP4/EDN1 addition to the culture medium is related to the formation of separate HAND2 and POU3F3 territories.

20) Lines 303-304; "It is unknown what mechanisms underlie this patterning event, but it is suggested that a kind of cell-sorting mechanism worked in it." This is too vague, please elaborate.

21) Please make sure that sample size is reported in the Figures or legends for every assay, including those reported as representative data.

22) Like I mentioned, the manuscript could elaborate more the strengths and potential applications of

this in vitro model. Moreover, the scRNAseq data could be better explored to bring out potential novel insights into e.g. early human craniofacial development, disorders affecting PA1, identification of novel putative specification factors, etc.

Reviewer #3 (Remarks to the Author):

This study by Yusuke Seto and coauthors “In vitro induction of patterned branchial arch-like aggregate from human pluripotent stem cells” introduces a novel method for generating aggregates containing a significant number of neural crest cells (NCCs) derived from human pluripotent stem cells (hPSCs). These aggregates can be differentiated into a brachial-arch like state. Unlike other protocols, this method does not require the addition of external small molecules, as the induction of NCCs is facilitated by aggregate-intrinsic WNT/BMP signals. This approach allows for the investigation of early signaling events involved in NCC specification under more physiological conditions. Single-cell RNA sequencing analysis of the aggregates revealed potential lineage bifurcation from a precursor state to NCCs, neural cells, and other cell types. Several candidate genes that play crucial roles in lineage specification were identified, including both known and uncharacterized genes, highlighting the need for further investigation. The long-term culture of these aggregates demonstrated their ability to differentiate into a maxillary arch-like state with the presence of bFGF, while collapse occurred without bFGF, indicating the importance of FGF signaling for mesenchymal cell survival. Moreover, the aggregates expressed marker genes associated with the mandibular arch when exposed to exogenous EDN1 and BMP4, suggesting the advantage of this in vitro model for studying craniofacial development. Interestingly, the temporal treatment with EDN1 and BMP4 resulted in the separation of HAND2+ cells and POU3F3+ cells within the aggregates, indicating a potential cell-sorting mechanism. Understanding this patterning event could shed light on in vivo facial patterning mechanisms. Further research is required to fully uncover the hidden mechanisms of early facial patterning using this in vitro model system.

Overall, this is a solid and much needed study, which can provide the community with the new advantageous model system to study facial, and more specifically, mandibular development. However, I have two large conceptual comments that the authors shall address prior to publishing:

1. One of the great values of this system would be the potential for the induction of jaw skeletogenesis, or at least the formation of skeletogenic condensations. It would be important to check if the authors can manage to push their organoid system into initiation of facial skeletogenesis. At least, the skeletogenic markers shall be studied in the current dataset. I would suggest to re-cluster the NCC-derived mesenchymal part of the dataset to study the developing heterogeneity of ectomesenchyme with better resolution. It will be important to achieve the induction of skeletal elements for publishing this story.

2. In Figure 8, the authors present the mixed mnd-mx identity of ectomesenchyme. However, it is important to resolve if this is a temporary state when the cells do not clearly resolve the borders between mx and mnd structures or it stays permanent. Does the similar state exist in real embryos where maxilla anatomically transits into mandible? I would have a more in depth look into natural/unnatural features of this state by comparing mouse embryos regional mnd mx signatures (identified from single cell seq and stainings) with those identified in human organoids in different

treatment regimes. I believe it is conceptually important to study this intermediate state and border refinement between mx and mnd identities, using single cell sequencing approach and other methods in mouse/chick embryos in comparison to human organoids.

REVIEWER COMMENTS (and POINT-BY-POINT RESPONSES)

Reviewer #1 (Remarks to the Author):

The Article “In vitro induction of patterned branchial arch-like aggregate from human pluripotent stem cells” by Yusuke Seto and Colleagues is very interesting and is likely to be useful for the NC community at large. They present sufficient evidence to suggest that cranial NCC can be induced in cultures of aggregated PSC using media devoid of exogenous WNT and BMP signal. The induction occurs in an efficient manner using 3 cell lines (2 hES and 1 iPSC) in just 5 days. The cultures do not render homogeneous population, with 100% of the cells undergoing NCC induction, however the numbers are robust enough. Furthermore the authors show that these cells acquire a cranial character belonging to the 1st branchial arch and that they can be directed to differentiate into mandibular, maxilar or both enabling multiple future explorations of the mechanisms at play. The figures are clear and provide strong evidence of the claims.

Response: We appreciate your review and constructive feedback. We performed several additional experiments and data analysis to respond your concerns. We believe these revisions according to your suggestions improved our manuscript. Please see point-by-point responses for each comment.

1st There are just a few suggestions and comments. The manuscript contains lots of typos and grammatical errors.

Response: We apologize for the inconvenience due to our grammatical issues. Our revised manuscript has been undergone proofreading by an experienced scientific editor.

2nd While at some point a reference is made to model organisms, but is missing in other places, for example, BMP and WNT have been shown to contribute to NC development in multiple organisms, statements along with adequate references are necessary.

Response: In accordance with your comment, we have added references regarding model animals. Newly added references were listed below.

2. Helms, J. A., Cordero, D. & Tapadia, M. D. New insights into craniofacial morphogenesis. *Development* **132**, 851–861 (2005).
8. Steventon, B., Carmona-Fontaine, C. & Mayor, R. Genetic network during neural crest induction: from cell specification to cell survival. *Semin Cell Dev Biol* **16**, 647–654 (2005).
11. Liu, W. *et al.* Threshold-specific requirements for Bmp4 in mandibular development. *Dev Biol* **283**, 282–293 (2005).
12. Vincentz, J. W. *et al.* Exclusion of Dlx5/6 expression from the distal-most mandibular arches enables BMP-mediated specification of the distal cap. *Proc Natl Acad Sci U S A* **113**, 7563–7568 (2016).
15. Simões-Costa, M. S., McKeown, S. J., Tan-Cabugao, J., Sauka-Spengler, T. & Bronner, M. E. Dynamic and differential regulation of stem cell factor FoxD3 in the neural crest is Encrypted in the genome. *PLoS Genet* **8**, e1003142 (2012).
16. Vöcking, O., Van Der Meulen, K., Patel, M. K. & Famulski, J. K. Zebrafish anterior segment mesenchyme progenitors are defined by function of tfap2a but not sox10. *Differentiation* **130**, 32–42 (2023).
17. Murdoch, B., DelConte, C. & García-Castro, M. I. Pax7 lineage contributions to the mammalian neural crest. *PLoS One* **7**, e41089 (2012).
20. Brugmann, S. A., Pandur, P. D., Kenyon, K. L., Pignoni, F. & Moody, S. A. Six1 promotes a placodal fate within the lateral neurogenic ectoderm by functioning as both a transcriptional activator and repressor. *Development* **131**, 5871–5881 (2004).
21. Kastriti, M. E. *et al.* Schwann cell precursors represent a neural crest-like state with biased multipotency. *EMBO J* **41**, e108780 (2022).
22. Kimura, J. *et al.* Emx2 and Pax6 function in cooperation with Otx2 and Otx1 to develop caudal forebrain primordium that includes future archipallium. *J Neurosci* **25**, 5097–5108 (2005).
23. Green, Y. S. & Vetter, M. L. EBF factors drive expression of multiple classes of target genes governing neuronal development. *Neural Dev* **6**, 19 (2011).
24. Betters, E., Liu, Y., Kjaeldgaard, A., Sundström, E. & García-Castro, M. I. Analysis of early human neural crest development. *Dev Biol* **344**, 578–592 (2010).
25. Jho, E. *et al.* Wnt/ β -Catenin/Tcf Signaling Induces the Transcription of Axin2, a Negative Regulator of the Signaling Pathway. *Mol Cell Biol* **22**, 1172–1183 (2002).
30. Hovland, A. S. *et al.* Pluripotency factors are repurposed to shape the epigenomic landscape of neural crest cells. *Dev Cell* **57**, 2257–2272.e5 (2022).
32. Roellig, D., Tan-Cabugao, J., Esaian, S. & Bronner, M. E. Dynamic transcriptional signature and cell fate analysis reveals plasticity of individual neural plate border cells. *Elife* **6**, e21620 (2017).

33. Thiery, A. P. *et al.* scRNA-sequencing in chick suggests a probabilistic model for cell fate allocation at the neural plate border. *Elife* **12**, e82717 (2023).
34. Williams, R. M., Lukoseviciute, M., Sauka-Spengler, T. & Bronner, M. E. Single-cell atlas of early chick development reveals gradual segregation of neural crest lineage from the neural plate border during neurulation. *Elife* **11**, e74464 (2022).
36. Matsuo, I., Kuratani, S., Kimura, C., Takeda, N. & Aizawa, S. Mouse *Otx2* functions in the formation and patterning of rostral head. *Genes Dev* **9**, 2646–2658 (1995).
37. Andoniadou, C. L. *et al.* Lack of the murine homeobox gene *Hesx1* leads to a posterior transformation of the anterior forebrain. *Development* **134**, 1499–1508 (2007).
38. Uehara, M. *et al.* CYP26A1 and CYP26C1 cooperatively regulate anterior-posterior patterning of the developing brain and the production of migratory cranial neural crest cells in the mouse. *Dev Biol* **302**, 399–411 (2007).
39. Williams, R. M. *et al.* Reconstruction of the Global Neural Crest Gene Regulatory Network In Vivo. *Dev Cell* **51**, 255-276.e7 (2019).
40. Litsiou, A., Hanson, S. & Streit, A. A balance of FGF, BMP and WNT signalling positions the future placode territory in the head. *Development* **132**, 4051–4062 (2005).
41. Simões-Costa, M., Tan-Cabugao, J., Antoshechkin, I., Sauka-Spengler, T. & Bronner, M. E. Transcriptome analysis reveals novel players in the cranial neural crest gene regulatory network. *Genome Res* **24**, 281–290 (2014).
42. Kim, Y.-K., Lee, H., Ismail, T., Kim, Y. & Lee, H.-S. *Dach1* regulates neural crest migration during embryonic development. *Biochem Biophys Res Commun* **527**, 896–901 (2020).
44. Collier, A. E. *et al.* GRHL2 and AP2a coordinate early surface ectoderm lineage commitment during development. *iScience* **26**, 106125 (2023).
46. Lopez-Pajares, V. *et al.* A LncRNA-MAF:MAFB transcription factor network regulates epidermal differentiation. *Dev Cell* **32**, 693–706 (2015).
48. Strobl-Mazzulla, P. H. & Bronner, M. E. Epithelial to mesenchymal transition: New and old insights from the classical neural crest model. *Seminars in Cancer Biology* **22**, 411–416 (2012).
49. Simões-Costa, M. & Bronner, M. E. Establishing neural crest identity: a gene regulatory recipe. *Development* **142**, 242–257 (2015).
55. Miller, C. T., Yelon, D., Stainier, D. Y. R. & Kimmel, C. B. Two *endothelin 1* effectors, *hand2* and *bapx1*, pattern ventral pharyngeal cartilage and the jaw joint. *Development* **130**, 1353–1365 (2003).

57. De Bono, C. *et al.* Single-cell transcriptomics uncovers a non-autonomous Tbx1-dependent genetic program controlling cardiac neural crest cell development. *Nat Commun* **14**, 1551 (2023).
59. Tribioli, C., Frasch, M. & Lufkin, T. Bapx1: an evolutionary conserved homologue of the *Drosophila* bagpipe homeobox gene is expressed in splanchnic mesoderm and the embryonic skeleton. *Mech Dev* **65**, 145–162 (1997).
60. Mori-Akiyama, Y., Akiyama, H., Rowitch, D. H. & de Crombrughe, B. Sox9 is required for determination of the chondrogenic cell lineage in the cranial neural crest. *Proc Natl Acad Sci U S A* **100**, 9360–9365 (2003).
61. Plouhinec, J.-L. *et al.* Pax3 and Zic1 trigger the early neural crest gene regulatory network by the direct activation of multiple key neural crest specifiers. *Dev Biol* **386**, 461–472 (2014).
62. Elms, P., Siggers, P., Napper, D., Greenfield, A. & Arkell, R. Zic2 is required for neural crest formation and hindbrain patterning during mouse development. *Dev Biol* **264**, 391–406 (2003).
63. Nakata, K., Nagai, T., Aruga, J. & Mikoshiba, K. *Xenopus* Zic3, a primary regulator both in neural and neural crest development. *Proc Natl Acad Sci U S A* **94**, 11980–11985 (1997).
64. Inoue, T. *et al.* Mouse Zic5 deficiency results in neural tube defects and hypoplasia of cephalic neural crest derivatives. *Dev Biol* **270**, 146–162 (2004).
65. Bhattacharya, D., Rothstein, M., Azambuja, A. P. & Simoes-Costa, M. Control of neural crest multipotency by Wnt signaling and the Lin28/let-7 axis. *Elife* **7**, e40556 (2018).
66. Piacentino, M. L., Hutchins, E. J. & Bronner, M. E. Essential function and targets of BMP signaling during midbrain neural crest delamination. *Dev Biol* **477**, 251–261 (2021).
69. Dash, S. & Trainor, P. A. The development, patterning and evolution of neural crest cell differentiation into cartilage and bone. *Bone* **137**, 115409 (2020).

3rd Interesting that WNT and BMP upregulation is shown at or after day 4, but according to the temporal inhibition, they trigger effects at earlier stages...this needs to be discussed, as irt does not match with the commitment suggested by authors.

Response: You have raised an important point. Considering that WNT/BMP inhibition from day 3 does not affect the induction of NCC-like cell, (1) WNT/BMP ligands expressed before day 3 must be important for NCC-like cell induction. Furthermore, (2) NCC-like cell induction itself should be independent of WNT/BMP signals after day 3 culture. To address these two points, (1) we performed additional real-time PCR analysis on *WNT* genes and found that a non-canonical WNT, *WNT7B*, was upregulated at day 1 (Fig. 3a in the revised manuscript). As for BMPs, we have already showed that *BMP4* was upregulated at day 1 in original manuscript (Fig. 3a in the revised manuscript). Furthermore, (2) we discussed possibility that autoregulation of NCC specifiers such as *PAX3* might contribute to induction of NCC-like cells in the absence of WNT/BMP signals from day 3 (pp.16-17, lines 375-388 in the revised manuscript), and additional analysis on scRNA-seq data supported that possibility (Supplementary Fig.16 in the revised manuscript). In the added text in Discussion, we have discussed the possible roles of WNT/BMP signals after day 3. Although it has not been shown experimentally, those signals might be involved in the multipotency and migratory potential of induced NCC-like cells, respectively, as indicated by chick studies (ref. 65 and 66 in the revised manuscript).

4th Today multiple human NC models also include TGFb inhibition, which here is not used but should be discussed

Response: In accordance with your suggestion, we have referred to TGFb inhibition by SB-431542 (p. 8, lines 182-187 and p. 17 lines 396-402 in the revised manuscript). We have also checked the effect of SB-431542 on our differentiation protocol and found that SB-431542 inhibits endoderm-like cell induction while enhances NCC-like cell induction (please see supplementary Fig. 5 in the revised manuscript).

The following notes direct to minor suggestions:

Page 4 line 93: The expression of both FoxD3 and Sox10 appears close to each other in model organisms and human embryos. Earlier NC markers include Pax3, Pax7, Sox9, etc.

Response: Taking your suggestion into account, (1) we added real-time PCR data of *PAX7* from day 0 to day 5 (Fig. 1c in the revised manuscript). (2) Furthermore, to characterize the expression pattern of neural crest markers, we performed immunohistochemistry of *PAX7*, *FOXD3*, and *SOX10* from day 3 to 5 and (Fig. 1d-e in the revised manuscript). Those markers were co-expressed in many cells but it seemed that *FOXD3* was downregulated in *SOX10*⁺ cells at day 5. Simões-Costa and colleagues showed that the expression timings of *FOXD3* and *SOX10* differed each other in chick embryos as *FOXD3* are expressed in pre-migratory NCCs prior to *SOX10* (Ref. 15 in the revised manuscript). Vöcking and colleagues showed co-expression of *FOXD3* and *SOX10* in newly-delaminating cNCCs in zebrafish embryos (Ref. 16 in the revised manuscript). In their paper, it seemed that *FOXD3* was downregulated in migratory NCCs, suggesting the downregulation of *FOXD3* along with the differentiation of NCCs. These data were consistent with the downregulation of *FOXD3* both in mRNA level and protein level at day 5 in the aggregates (Fig. 1c-d in the revised manuscript). Expression pattern of *PAX7* was broader than that of *FOXD3* and *SOX10* as shown in Fig. 1d-e in revised manuscript. This result was consistent with that *PAX7* was expressed not only NCCs but also dorsal neuroepithelium *in vivo* (Ref. 17 in the revised manuscript). We have added related text in pp. 4-5, lines 96-109 in the revised manuscript.

Page 5 Line 97...”neural cells express p75 at high and low levels, respectively 3”. **Bettors et al (Developmental Biology 344 (2010) 578–592) showed in 2010 the lack of NC specificity of P75 for neural crest cells in human embryos, and thus this seem appropriate to include this earlier reference.**

Response: We agree with the relevance of suggested reference, and have added it to the text (ref. 24 at p.6, line 127 in the revised manuscript). Furthermore, we have addressed whether p75 could be used as specific markers for NCC-like cells in day-5 aggregates (see Supplementary Fig. 3). The results of additional experiments suggested that p75 was specific to *SOX10*⁺ NCC-like cells in the aggregates.

Lines 98-99: Supp Fig1 D does not show P75 analyses.

Response: Your comment is correct. We have added immunohistochemistry of p75 and *SOX10* and flow cytometry analysis with anti-p75 antibody for hPSC lines as Supplementary Fig. 3 in the revised manuscript instead of Supplementary Fig. 1D in the original manuscript.

Line 113-114: ... “confirmed the presence of those types of cells in the aggregates by immunohistochemistry (Fig. 1b, 2e, f).” No immunos corresponding to this panels or stainings found. Only Six1 and Isl 1/2.

Response: Your comment is correct. In revised version of manuscript, we tried more detailed characterization of clusters in day-5 aggregates (Fig.2) and performed immunohistochemistry to identified those clusters (please see Supplementary Fig.2). Corresponding text is pp. 5-6, lines 110-126 in the revised manuscript.

Lines 115-116: “Unlike other NCC induction methods from hPSC, extrinsic manipulation of WNT/BMP signals was not necessary in our method 5. Include Hackland et al 2019 (Stem Cell Reports j Vol. 12 j 920–933) whci modulates BMP, and WNT. Leung et all, relies on WNT only.

Response: We appreciate your comment. We have added Hackland et al 2019 as a reference (ref. 7 in the revised manuscript)

Page 9, lines 206-207: “POU3F3 which is strongly expressed in the maxillary arch 34, but they did not express DLX5 which is expressed in the mandibular arch (Fig. 6c).” Adding the fluorescence from the supplemental figure showing the negative expression by Immuno is relevant, particularly given that the levels of the Q-RT-PCR are so close.

Response: In accordance with your suggestion, we have added immunohistochemistry of DLX5 in Day-4 and Day-12 aggregates (please see Supplementary Fig. 13). DLX5 was expressed only in epithelial cells in Day-4 aggregates and Day-12 non-treated/bFGF-treated aggregates but it was expressed also in mesenchymal cells in Day-12 EDN1/BMP4-treated aggregates (positive control).

Page 12 Lines 206-207: “POU3F3 which is strongly expressed in the maxillary arch 34, but they did not express DLX5 which is expressed in the mandibular arch (Fig. 6c).” This tentative sentence should be moved to the discussion and should be followed by a more informative speculation.

Response: We are sorry if we are not correct but our understanding is that you might comment on

the idea of domain formation in the aggregates cell sorting by CDH11. We moved description about it to the discussion and append more detailed description. We have also described the idea that chemotaxis might have roles in gathering same type of cells in the aggregates. Corresponding text is pp. 18-19, lines 424-437 in the revised manuscript.

Thank you once again for your valuable comments. We feel that newly added data and discussion according to your comments have strengthened the scientific evidence of our claims. We believe this *in vitro* model would facilitate the research of neural crest cells especially in human.

Reviewer #2 (Remarks to the Author):

Seto et al reports a novel in human vitro model that generates 3-dimensional aggregates containing neural crest cells (NCCs) from pluripotent stem cells. Temporal scRNAseq analysis revealed that NCCs were specified from an intermediate neural plate border (NPB) stage followed by an EMT event. Further, prolonged cultivation under exposure to EDN1/BMP4 treatments resulted in acquisition of pharyngeal arch (PA) identity, and both mandibular and maxillary arches (derived from the first PA) within the aggregates. These observations suggest recapitulation of key developmental events, providing opportunities for the investigation of human NCC development, including cell patterning and acquisition of positional identity. In my opinion, this is relevant for 3 reasons: 1) early specification and patterning events during craniofacial development are complex processes that still need clarification; 2) the strategy could be used not only to bring insights into early human NCC development (for which models are scarce), but also into the etiology of several congenital disorders, which often present with mandibular and maxillary defects; 3) the simplicity (and probable cost effectiveness) of the model makes it quite accessible.

This work has great potential to contribute to the field, but it is lacking in some aspects in its current form. Such contributions and impact seem a bit underappreciated throughout the text, and perhaps a more solid demonstration of its potential to bring new insights could significantly improve the manuscript. Please find below some issues I think should be resolved:

Response: We appreciate your insightful feedback. We performed several additional experiments and data analysis to respond your concerns. We have gained more insights into the induction of NCCs in hPSC culture from those experimental results and analysis. We believe that these revisions improved our manuscript. Please see point-by-point responses for each comment.

1) General writing – authors should consider revising the manuscript to resolve grammatical/stylistic issues. Some sections are not clear, for example:

a) Abstract, line 36 - I presume you mean “mean specification and patterning events”

b) Introduction, lines 39-41 – This section sounds odd, especially the terms used (e.g. facial identifiers)

c) Introduction, line 44 - “The NCCs are transient cell population which arises from neural plate border

(NPB). They migrate from dorsal their origin to ventral craniofacial primordium”. Did you mean “They migrate from their dorsal origin to ventral craniofacial primordia”?

d) Introduction, line 62 – “...their capability to directionally differentiate into NCC”. Do authors mean “directed differentiation” or “direct differentiation”?

Response: We appreciate your comment and the correspondence is as follows.

a) Considering your comment, we rewrite the text. Passages corresponding are in p. 2, lines 35-37 in the revised manuscript.

b) We removed the corresponding passages and simplified the text (p. 3, lines 50-51 in the revised manuscript).

c) We have rewritten as you suggested.

d) We have rewritten to “directed differentiation”.

2) Introduction, lines 52-56 - This section seems oversimplified. Reference 2 reports that post migratory NCC movements within the PA are involved in a dynamic cell fate specification process, which takes place via sequential fate bifurcations as opposed to specification in discrete compartments of the PA, as previously thought. Further, these observations were made in postmigratory NCCs, so stating that 'migration of NCCs is involved in that process' is misleading as it may refer to migratory NCCs.

Response: We agree with your comment and have written description about that reference (ref. 3 in the revised manuscript). Corresponding text is p. 3, lines 61-64 in the revised manuscript.

3) Results, lines 97-99 - “On day 5, approximately 70% of cells in the aggregates were p75 bright, although the percentage varied among cell lines (Fig. 1d and Supp. Fig. 1D).”

Variation in p75 dim/bright in flow cytometry can be indicative of variation in the proportion of NCC vs neural cells, but the immunostaining of cell aggregates (Suppl. Fig. 1D) show CDH1+ clusters (neither a neural or NCC marker). Please make a distinction between these two lines of evidence to support your conclusion here.

Response: We apologize for confusing description in corresponding passages. First of all, we have misinterpreted the result of a related reference (Fukuta et al 2014). In that paper, they have shown the enrichment of NCCs in p75^{bright(high)} fraction but not mentioned that p75^{dim(low)} cells corresponded to neural cells. It seemed that p75^{low} fraction include not only neural cells but also other types of cells. Thus, we have rewritten the description about p75 to focus on only p75^{high}

cells. Accordingly, we have deleted Suppl. Fig. 1D in previous version of the manuscript and added relevant data as Supplementary Fig. 3 in the revised manuscript. We have performed immunohistochemistry of SOX10 and p75 (and PAX6) to show the specificity of p75 expression to SOX10⁺ cells (Supplementary Fig. 3b, c in the revised manuscript). Time course analysis (Supplementary Fig. 3c in the revised manuscript) showed matched expression of SOX10 and p75 in the aggregates. Expression of p75 was almost specific to SOX10⁺ cells in the aggregates derived from other hPSC lines, KthES11 and 253G1. To show the variation in p75 low/high, we also added histogram of flowcytometry analysis as Supplementary Fig. 3d. Corresponding text is p. 6, lines 127-134 in the revised manuscript.

4) Please specify which cell line(s) was used in scRNAseq assays. Did you assess p75 expression in your scRNAseq data (Fig. 2b)? This could indicate whether flow cytometry for p75 alone has sufficient resolution to estimate the % of NCCs. p75 is a membrane receptor, so this could be informative to researchers attempting to sort NCCs using this experimental framework.

Response: We agree with your comment because this is a very important point. We used KhES-1-derived aggregates in scRNA-seq experiment. We have added that description in Results (p. 5, line 111 in the revised manuscript) and Methods (p. 23, line 530 in the revised manuscript). Furthermore, according to your comment, we added feature plot of p75 as Supplementary Fig. 3a to show the specific expression of p75 in SOX10⁺ NCC-like cells. Immunostaining of p75 and SOX10 (Supplementary Fig. 3b, c) in the revised manuscript also supported that p75 expression is almost specific to SOX10⁺ NCC-like cells.

5) Line 106, “were classified as SOX10+ NCCs (Fig. 2b)”. Fig. 2c must be referenced here as well.

Response: We agree with you and incorporated this suggestion as follows: “Clusters 0, 1, 3, 4, 5, 6 and 7 represented *SOX10*⁺ NCC-like cells, accounting for 71.9% of the total cells (Fig. 2c, d)” (p.5 lines 113-114 in the revised manuscript). Fig. 2d in revised manuscript correspond to Fig. 2b in original.

6) Fig 3b: The layout makes it difficult to read panel b.

Response: According to your comment, we modified the layout of Fig. 3b.

7) Lines 122-124, referring solely to SOX10 expression upon WNTi/BMPi treatment: This is also valid for PAX6, SIX1 and EYA1.

Response: We agree with your comment. We have written about the expression of *PAX6*, *SIX1* and *EYA1* upon WNTi/BMPi treatment (p. 7, lines 158-166 in the revised manuscript).

8) Lines 124 – 127: Please specify to which treatment day you are referring (for CDH1 EYA1 and PAX6)

Response: According to your comment, we have specified treatment day in revised manuscript (p. 7, lines 158-166 in the revised manuscript).

9) Fig 3d: Please provide magnifications

Response: According to your comment, we have added magnifications for Fig. 3d.

10) Lines 131-133 “We found that, a lot of cells in SIX1+ PPE ...expressed PAX6” (Fig. 3e): Don't you mean SIX1+/CDH1+? CDH1 is expressed in the PPE and magnifications in Fig. 3e show CDH1+ regions. Please remove comma after “We found that”.

Response: Your comment is correct. We have rewritten as follows and removed comma:

In original manuscript (p. 6, lines 131-133)

“We found that, a lot of cells in SIX1⁺ PPE in the WNT inhibitor-treated aggregates expressed PAX6, while PPE in the control aggregates did not (Fig. 3e).”

In revised manuscript (p. 8, lines 174-176)

“We confirmed that there were SIX1⁺/CDH1⁺ cells that did not express PAX6 in non-treated aggregates, while many SIX1⁺/CDH1⁺ cells co-expressed PAX6 in IWP-2-treated aggregates, suggesting they became anterior PPE-like cells (Fig. 3e).”

11) Figure 4: In Fig. 4b, IRX1 and SP5 do not seem specific to the NPB (clusters 5 and 10). The NPB expresses a variety of markers of all ectodermal derivatives, so I understand it might be difficult to bring forward specific expression patterns. Still, it is known that the NPB expresses neural and non neural transcription factors with some level of co-

localization (e.g. *Tfap2a/Pax7*; *Tfap2a/Sox2*, etc). What other NPB markers are expressed in these clusters? Did you observe these co-expression patterns? It would greatly add to the manuscript If authors provided a more in-depth analysis and characterisation of the NPB clusters, as fate specification within the NPB is an active area of research with very little data on human cells.

Check e.g. Williams et al., 2022 and Roellig et al., 2017:

<https://elifesciences.org/articles/21620>

<https://elifesciences.org/articles/74464>

Besides clusters 5,10, 11, what other cell populations were identified here? Can you observe cell fate trajectories from NPB to NCC, PPE and Neural Plate lineages in any specific order? Which expression patterns characterize these trajectories? Can novel putative specification factors be identified under this approach?

Response: Taking your comment in account, we performed additional analysis of scRNA-seq.

(1) Analysis of co-expression of *SOX2* and *TFAP2A* (Supplementary Fig. 7)

According to Roellig and colleagues (which you mentioned above), *SOX2* and *TFAP2A* are co-expressed in NPB. Thus, we have analyzed the expression of *SOX2* and *TFAP2A* in each data (day 2 to day 5) and checked the co-expression of those genes (Supplementary Fig. 7a). We found that *SOX2* and *TFAP2A* were co-expressed in a subset of cells at every stage tested. Furthermore, we analyzed that co-expression by IHC of *SOX2*, *TFAP2A* (and *PAX7*) (Supplementary Fig. 7b). We observed co-expression of *SOX2* and *TFAP2A* at every stage, consistent with scRNA-seq analysis. Interestingly, the expression of *SOX2* was downregulated in most *TFAP2A*⁺ cells from day 4. In those *SOX2*-downregulated cells, the expression of *PAX7* was upregulated, indicating lineage bifurcation have occurred there. Corresponding text is pp. 9-10, lines 209-217 in the revised manuscript.

(2) Further characterization of NPB-like cell clusters (Supplementary Fig. 8 and 9)

To characterize NPB-like cells, we have extracted NPB-like cell clusters from merged scRNA-seq data and re-clustering has been performed (Supplementary Fig. 8a). Although it was difficult to find a single marker gene which could specify a single cluster alone, we found that several genes expressed in distinct manners in NPB-like cells (Supplementary Fig. 8b, c and Supplementary Fig. 9), suggesting a variety of cell state in NPB-like cells. Corresponding text is pp. 10-11, lines 218-245 in the revised manuscript.

(3) Trajectory analysis of NPB-like cells (Supplementary Fig. 10)

By using R packages slingshot and tradeSeq, we have performed trajectory analysis of NPB-like cells and identified three trajectories corresponding to NCC-like, neuroepithelial cell-like, NNE/PPE-like lineages. By this analysis, several genes known for their roles in the development (for example, *GRHL3*, *ETSI*, *MYCN* etc.) were found to be modulated during differentiation. Furthermore, we also identified several functionally-unknown genes (for example, *ELF3* and *KLF6* for NNE/PPE lineage and *EBF1* and *SEZ6* for NCC lineage), suggesting our *in vitro* model might be able to give insight into the developmental mechanisms of NPB development. Corresponding text is pp. 11, lines 246-254 in the revised manuscript.

12) Fig 4c: what is being indicated by the colours?

Response: RNA velocity analysis was done with scVelo on python and the colours used for UMAP of Fig. 4c in original manuscript were default setting. In revised manuscript, we have matched the colours of that UMAP to others. In revised manuscript, Fig. 4h corresponds to Fig. 4c in original manuscript.

13) Line 162: Please specify which GO categories were enriched, and in which cluster(s)/differentiation days. It would be nice if you could show that the time-series analysis of EMT markers replicate some or all of the temporal expression patterns observed in vivo.

Response: We performed GO analysis using clusters in merged scRNA-seq data (please see Supplementary Fig. 11a in the revised manuscript). We have also specified enriched GO categories and clusters in main text (pp. 11-12, lines 262-265 in the revised manuscript). To show the temporal expression patterns of EMT-related genes, we have made heatmap of those genes based on pseudotime analysis (Fig. 5a in the revised manuscript) and described the similarity to *in vivo* development in main text (p. 12, lines 265-269 in the revised manuscript).

14) Fig 7d and others; Multiple comparisons in the manuscript: consider using ANOVA +posthoc tests.

Response: According to your suggestion, we have performed ANOVA +posthoc tests for multiple comparisons. Responding graphs are as follows: Fig. 7d, f, Fig. 8d, Supplementary Fig. 15c in the revised manuscript.

15) Lines 266 – 274: This part sounds a bit vague. Please provide a reference for cadherin-mediated cell sorting and briefly explain how your data may recapitulate this process. Also, how is this related to the higher *in vivo* expression of CDH11 in the mandibular arch? Please clarify the conceptual basis for this statement.

Response: We apologize for the vague description on the idea of a possible mechanism for separation of HAND2⁺ cells and POU3F3⁺ cells in the aggregates. Actually, Kimura et al., 1995 (ref. 69 in the revised manuscript) showed segregation of Cdh11⁺ cells and Cdh11⁻ cells by *in vitro* mixed cell aggregation assay using mesenchymal cells dissociated from embryonic limb which contained Cdh11⁺ cells. We suppose expression of CDH11 may facilitate the separation of HAND2⁺ MN-like cells from POU3F3⁺ MX-like cells through sorting of CDH11⁺/DLX5⁺ MN-like cells (including future HAND2⁺ cells) from other CDH11⁻ cells (including future POU3F3⁺ cells) at early phase of differentiation in the aggregates. We also suppose that there may be other cell adhesion molecules differentially expressed between MX-like cells and MN-like cells and CDH11 may be just a one of them. We presumed that total difference in the expression of cell binding molecules might be important for cell segregation in MN because there has been no literature which described abnormalities in facial morphology in *Cdh11*-KO animals. Actually, we have no our own experimental data for cell sorting by CDH11 in the aggregates so we moved the description about CDH11 from Results to Discussion. Related texts were p. 18, lines 424-428 and pp. 18-19, lines 431-437 in the revised manuscript.

16) Suppl. Fig. 3: Please include magnifications showing DLX5⁺/CDH11⁺ cells. These aggregates were exposed to which bFGF/EDN1/BMP4 treatment regime?

Response: According to your comment, we added magnifications (please see Supplementary Fig. 17b in revised manuscript). At day 7 (early timing of differentiation into branchial arch mesenchyme-like cells), CDH11 was expressed only around the inner core of the aggregates in condition 1 indicated in Fig. 8b (bFGF-treated) suggesting CDH11 might expressed around the NPB-like or NCC-like cells undergoing EMT-like processes while CDH11 was expressed in DLX5⁺ MN-like cells in EDN1/BMP4-treated aggregates (condition 2 and 3 in Fig. 8b in revised manuscript). At day 7, condition 2 and 3 is same configuration so we do not distinguish them. We added the information of conditions (treatment regime) used for culture to the figure legend (Supplementary Fig. 17 in the revised manuscript).

17) Discussion, lines 278-280: The efficiency of NCC generation achieved by the protocol was 45.4-76.5% considering only p75 expression, and assuming p75 alone is sufficient to identify NCCs (which has yet to be demonstrated); and 70.3% according to scRNAseq data. The reported efficiency seems to lie below the latest methods for NCC generation, at least in 2D systems. Please clarify which work is being referenced here. Moreover, since this is a novel NCC induction strategy, their ability to terminally differentiate into NCC derivatives should be demonstrated.

Response: (1) In Leung et al., 2016 (Ref. 5 in the revised manuscript), they said that $63.1 \pm 9.6\%$ of cells expressed SOX10 in their method. In our method, KhES-1 and KthES11 ESC lines showed $76.5 \pm 3.1\%$ and $70.9 \pm 2.1\%$ of cells were p75^{high} cells so we described as “In this method, NCCs can be induced within 5 days with an efficiency comparable to that of a recently reported, highly efficient protocol for NCC induction from hPSCs” in original manuscript. But as you mentioned above, other protocols (for example, Menendez et al., 2011 (Ref. 6 in the revised manuscript)) showed higher efficiency. Take this into consideration, we have decided to remove that description from the manuscript. Instead, we have just described the induction efficiency in p. 16 lines 369-371. (2) In accordance with your comment, we have performed additional experiment to test the ability of induced cells to differentiate into multiple cell types. We have confirmed differentiation into neural (TUJ1⁺/SIX1⁺) and glial (GFAP⁺/SOX10⁺) cells, melanoblasts (MITF⁺), smooth muscle cells (SMA⁺), osteoblasts (indicated by Alizarin Red staining), chondrocytes (indicated by Alcian Blue staining), adipocytes (indicated by Oil Red O staining). Corresponding text is p. 6, lines 135-137 in the revised manuscript and corresponding figure is Supplementary Fig. 4.

18) Lines 282-283: Please, elaborate on why the reported in vitro model is more 'beneficial' for investigating early NCC specification. What are the advantages over current methods? How does it compare to current in vitro/in vivo models?

Response: We apologize for the vague description in lines 282-283 in the original manuscript. We feel “beneficial” is not appropriate now. First of all, we agree with your comment that fate specification in NPB is active research area. We assume our method is “suitable” for analyzing

the mechanism of cell fate specification within NPB because multiple ectodermal lineages arise from NPB-like state within our aggregate as suggested by scRNA-seq analysis (Fig. 4h). Furthermore, we suppose that our *in vitro* model may have advantage to analyze the role of environmental signals because our model does not need any extrinsic signal regulators such as CHIR-99021. Our model would also be suitable for several methodologies such as live imaging using reporter cell lines, phenotype analysis using knockout or transgenic cell lines, CRISPR screening, and so on. We apologize that we could not write the advantages of our method in detail because of word limit. In revised manuscript, we use “suitable” instead of “beneficial”. Corresponding text is in p. 16, lines 368-374 in the revised manuscript.

19) Lines 300-303: It is not completely clear how the method of BMP4/EDN1 addition to the culture medium is related to the formation of separate HAND2 and POU3F3 territories.

Response: Your comment is correct. We could not reveal the mechanisms underlying that separation in this study and we think it is an important future challenge. We suppose that biased localization of DLX5⁺ cells in BMP4/EDN1-treated aggregates may be a clue (please see Supplementary Fig. 17). Although BMP4 and EDN1 were bath-applied in our method, such biased localization of DLX5⁺ cells were fairly common. We suppose that self-assembly of DLX5⁺ cells within an aggregate may be a potential mechanism of such biased localization. We also suppose directional migration and homophilic adhesion may be potential underlying events of self-assembly. In BMP4/EDN1-treated WNT5A is upregulated as shown in Fig. 7f. WNT5A has been known to act as chemoattractant in human neutrophils (ref. 68 in the revised manuscript). If DLX5⁺ cells expressed corresponding receptor selectively, WNT5A would be work to gather cells within the aggregate. Although we have not analyzed which cells expressed *WNT5A* and its receptors, *Wnt5a* is expressed in MN specifically *in vivo* (Supplementary Fig. 15b). Thus, WNT5A-dependent chemotaxis might work to gather DLX5⁺ MN-like cells in the aggregates if DLX5⁺ cells express specific receptors needed for chemotactic migration. As for homophilic adhesion of cells, as an example, we show that the expression of CDH11 was upregulated in DLX5⁺ MN-like cells selectively. During the development of mouse embryo, the expression of CDH11 is expressed during the early morphogenesis of MX, MN and limb bud transiently (ref. 69 in the revised manuscript). *Cdh11* showed homophilic binding activity and mixed cell aggregation using limb mesenchyme also showed sorting of *Cdh11*⁺ cells from *Cdh11*⁻ cells. We want to emphasize that upregulation of CDH11 in MN-like cells at early differentiation timing such as day 7 is just an example of changes in cell adhesion property. Although we could not

analyze yet, there may be other factors that make difference in the cell adhesion property between MN-like and MX-like cells in the aggregate. We have to analyze those points in future study to reveal the mechanism of separation of two types of cells and its relevance to *in vivo* development. Text corresponding to those description is in pp. 18-19, lines 419-437 in the revised manuscript.

20) Lines 303-304; “It is unknown what mechanisms underlie this patterning event, but it is suggested that a kind of cell-sorting mechanism worked in it.” This is too vague, please elaborate.

Response: We agree with you and have rewritten the text as shown in Response to comment 19.

21) Please make sure that sample size is reported in the Figures or legends for every assay, including those reported as representative data.

Response: We performed at least three independent culture for IHC and other staining experiment. We have added that description in Materials and Methods (pp. 22, lines 507-508 in the revised manuscript). As for real-time PCR analysis and flowcytometry analysis, sample size has been reported in legends.

22) Like I mentioned, the manuscript could elaborate more the strengths and potential applications of this in vitro model. Moreover, the scRNAseq data could be better explored to bring out potential novel insights into e.g. early human craniofacial development, disorders affecting PA1, identification of novel putative specification factors, etc.

Response: We sincerely appreciate your suggestions especially for in-depth analysis of scRNA-seq data which have given us more insight into the developmental process of the aggregate. We believe that we could elaborate our manuscript by incorporating your feedback. Thank you for taking the time to help us improve this paper. We also hope that our response satisfactorily address concerns you have noted.

Reviewer #3 (Remarks to the Author):

This study by Yusuke Seto and coauthors “In vitro induction of patterned branchial arch-like aggregate from human pluripotent stem cells” introduces a novel method for generating aggregates containing a significant number of neural crest cells (NCCs) derived from human pluripotent stem cells (hPSCs). These aggregates can be differentiated into a branchial-arch like state. Unlike other protocols, this method does not require the addition of external small molecules, as the induction of NCCs is facilitated by aggregate-intrinsic WNT/BMP signals. This approach allows for the investigation of early signaling events involved in NCC specification under more physiological conditions. Single-cell RNA sequencing analysis of the aggregates revealed potential lineage bifurcation from a precursor state to NCCs, neural cells, and other cell types. Several candidate genes that play crucial roles in lineage specification were identified, including both known and uncharacterized genes, highlighting the need for further investigation. The long-term culture of these aggregates demonstrated their ability to differentiate into a maxillary arch-like state with the presence of bFGF, while collapse occurred without bFGF, indicating the importance of FGF signaling for mesenchymal cell survival. Moreover, the aggregates expressed marker genes associated with the mandibular arch when exposed to exogenous EDN1 and BMP4, suggesting the advantage of this in vitro model for studying craniofacial development. Interestingly, the temporal treatment with EDN1 and BMP4 resulted in the separation of HAND2+ cells and POU3F3+ cells within the aggregates, indicating a potential cell-sorting mechanism. Understanding this patterning event could shed light on in vivo facial patterning mechanisms. Further research is required to fully uncover the hidden mechanisms of early facial patterning using this in vitro model system.

Overall, this is a solid and much needed study, which can provide the community with the new advantageous model system to study facial, and more specifically, mandibular development. However, I have two large conceptual comments that the authors shall address prior to publishing:

Response: We appreciate your review and intriguing feedback. We performed several additional experiments and data analysis to respond your comment. We believe that revisions according to your suggestions has revealed new uses for this *in vitro* model. Please see point-by-point responses for each comment.

1. One of the great values of this system would be the potential for the induction of jaw

skeletogenesis, or at least the formation of skeletogenic condensations. It would be important to check if the authors can manage to push their organoid system into initiation of facial skeletogenesis. At least, the skeletogenic markers shall be studied in the current dataset. I would suggest to re-cluster the NCC-derived mesenchymal part of the dataset to study the developing heterogeneity of ectomesenchyme with better resolution. It will be important to achieve the induction of skeletal elements for publishing this story.

Response: Your suggestion is very interesting for us. Unfortunately, ectomesenchyme population which expressed *TWIST1* etc was not included in our scRNA-seq dataset (day2 to day 5). Thus, we performed additional experiment to analyze differentiation into osteogenic lineage in the aggregates (Fig. 9. Corresponding text is pp. 15-16 lines 358-365 in the revised manuscript). We cultured the aggregates which was treated with EDN1/BMP4 temporarily for longer period than day 21 in the presence of bFGF. We analyzed the expression of marker for osteogenic lineage at day 35 and found that there were RUNX2⁺ cells in the aggregates. Furthermore, subset of RUNX2⁺ cells also expressed SP7/OSTERIX indicating the differentiation into osteogenic lineage. In addition to markers for osteogenic lineage, we analyzed the expression of markers related to chondrogenesis. Interestingly, we found that there were cell condensates formed by SOX9⁺ cells. Some cells also expressed NKX3.2/BAPX1 in those condensates, which were expressed in Meckel's cartilage in mouse embryos. Those results suggested chondrogenesis in the aggregates. In this experiment, we added only bFGF to the medium after day14 so differentiation into osteogenic/chondrogenic lineage have occurred spontaneously in the aggregates, raising the possibility that this *in vitro* model could also be used as a model for fate specification of osteogenic/chondrogenic lineage in facial primordium. Although SP7 was expressed in day-35 aggregates, we could not observe apparent signals of Alizarin Red S staining even when we added β-Glycerophosphate and ascorbic acid ± dexamethasone in medium suggesting that much longer culture period or optimization of culture medium may be needed to induce osteogenesis in the aggregates as a next challenge. We believe such a challenge will help the analysis of the mechanism regulating facial bone/cartilage development which is also active area of research.

2. In Figure 8, the authors present the mixed mnd-mx identity of ectomesenchyme. However, it is important to resolve if this is a temporary state when the cells do not clearly resolve the borders between mx and mnd structures or it stays permanent. Does the similar state exist in real embryos where maxilla anatomically transits into mandible? I would have a more in depth look into natural/unnatural features of this state by comparing mouse embryos regional mnd mx signatures (identified from single cell seq and stainings) with

those identified in human organoids in different treatment regimes. I believe it is conceptually important to study this intermediate state and border refinement between mx and mnd identities, using single cell sequencing approach and other methods in mouse/chick embryos in comparison to human organoids.

Response: First of all, we have to sincerely apologize for the misleading schematic diagram shown as Fig. 8e in the original manuscript. It is not that we wanted to mention that there were cells with mixed maxillary (MX)-mandibular (MN) identity (which means co-expression of MX marker and MN marker) in the aggregates treated with EDN1 and BMP4 temporarily, we just wanted to say that there were both maxillary arch (MX)-like cells and mandibular arch (MN)-like cells in the aggregates cultured in that condition. Actually, we did not observe any cells co-expressing MX marker *POU3F3* and MN marker *HAND2*. To clarify this, we added the magnification of immunostaining of *POU3F3* and *HAND2* for the temporary-treated aggregate (please see Supplementary Fig.15f in the revised manuscript). We also fixed the schematic diagram in Fig.8e in the revised manuscript so as not to mislead the readers.

Even considering the above, we still feel that your suggestion about comparison with model animals is important. Thus, we analyzed scRNA-seq data of E9.5 mouse branchial arch reported in De Bono et al., 2023 (Ref. 58. Corresponding text is p. 15, lines 345-357 in the revised manuscript). This data included not only first branchial arch but also other branchial arches so we extracted HOX-negative cells from the dataset and re-clustering cells (Supplementary Fig. 15a in the revised manuscript). As a result, we have obtained MX and MN clusters determined by the expression of known markers (for example, *POU3F1*, *DLX5*, and *HAND2*). We have also obtained new markers such as *EBF1*, *POU3F1* for MX and *DLK1* for MN (Supplementary Fig. 15b in the revised manuscript). We have performed real-time PCR analysis of day-21 aggregates for those genes (Supplementary Fig. 15c in the revised manuscript). In the aggregates cultured with only bFGF (Cond.1: MX-like aggregates), MX markers *EBF1* and *POU3F1* were expressed at high levels while the expression level of *DLK1*, a MN marker, was low. On the other hand, Expression levels of MX markers were low in the aggregates cultured with EDN1/BMP4 by day 21 (Cond. 3: MN-like aggregates) and these aggregates expressed a MN marker *DLK1* at high level. In temporary-treated aggregates (Cond. 2: MX+MN-like aggregates), MX markers and MN markers were expressed at intermediate levels. Those results suggested the similarity in the gene expression between mouse first branchial arch and our *in vitro* model. Furthermore, we have shown the mutual expression of *Pou3f3* and *Hand2* in the first branchial arch at this stage (E9.5) (Supplementary Fig. 15d in the revised manuscript). We also have checked the expression of those markers by immunohistochemistry (Supplementary Fig. 15e

in the revised manuscript). Similarly to the result obtained from immunohistochemistry of the aggregates, there was no overlap of *Pou3f3* and *Hand2* in first branchial arch. Again, those results suggested that our *in vitro* model recapitulates characteristics of first branchial arch to some extent. Although we could not find cells expressing both POU3F3 and HAND2 simultaneously, it does not mean that intermediate state between MX and MN does not exist. HAND2 is expressed from day 7 in the aggregates while expression of POU3F3 is upregulated after day 12. So POU3F3 may be not a good marker for early MX-like cells. Exploration of early marker for MX-like cells will facilitate further research on fate bifurcation between MX-like cells and MN-like cells and border refinement between MX and MN. Our *in vitro* model would be suitable for analyzing such an early process during the development of first branchial arch.

Thank you for taking the time and effort in reviewing our manuscript. We believe these revisions according to your comments have shown new possibilities of our *in vitro* model. We hope this study will contribute to the further development of this research area.

Figures and Tables:

To improve the manuscript according to the reviewers' comments, we have changed figure composition as described below:

Previous Fig. 1b has been modified and renamed as Fig. 1c.

Previous Fig. 1c has been renamed as Fig. 1b.

New data has been inserted as Fig. 1d, e.

Previous Fig. 1d has been moved to Supplementary Fig. 3d.

Previous Fig. 2b has been modified and renamed as Fig. 2d, h, i, j

Previous Fig. 2d has been modified and renamed as Fig. 2b.

Previous Fig. 2d, e has been modified and renamed as Fig. 2c, b, respectively.

New graph has been inserted as Fig. 2g.

Fig. 3a has been modified.

Layout of Fig. 3b has been modified according to reviewer's comment.

Fig. 3d, e has been modified according to reviewer's comment.

Previous Fig. 4b has been modified and renamed as Fig. 4b-g.

Previous Fig. 5a has been removed.

Previous Fig. 5b has been removed.

New data has been inserted as Fig. 5a.

Previous Fig. 5c has been renamed as Fig. 5b.

Previous Fig. 5d has been renamed as Fig. 5c.

Previous Fig. 5e has been modified and renamed as Fig. 5d.

Previous Fig. 6a has been moved to Supplementary Fig. 8b except for *HOXA2*.

Feature plot of *HOXA2* in previous Fig. 6a has been moved to Supplementary Fig. 12.

Previous Fig. 6b has been modified and renamed as Fig. 6a.

Previous Fig. 6c has been renamed as Fig. 6b.

Previous Fig. 6d has been renamed as Fig. 6c.

Previous Fig. 6e has been renamed as Fig. 6d.

Previous Fig. 6f has been renamed as Fig. 6e.

Previous Fig. 6g has been renamed as Fig. 6f.

Schematic diagram in Fig. 8e has been modified.

Fig. 9 has been inserted.

Previous Supplementary Fig. 1D has been removed.

Previous Supplementary Fig. 2 has modified and renamed as Supplementary Fig. 11.

Previous Supplementary Fig. 3 has modified and moved to Supplementary Fig. 17.

New data have been inserted as Supplementary Figures.

Previous Supplementary Tables 1 have been modified.

Previous Supplementary Tables 2 have been modified and renamed as Supplementary Tables 4.

New Supplementary tables have been inserted as Supplementary Tables 2 and 3.

In addition to those changes, appearance of scRNA-seq data have changed from original manuscript. This is due to technical reason (change in version of R package Seurat).

REVIEWERS' COMMENTS

Reviewer #1 (Remarks to the Author):

The reviewed manuscript incorporates recommendations and it is much improved. Really only minor suggestions from this reviewer for consideration. The Introduction, first paragraph:
Odd description of germ layers and NC contributions to the face, as no endoderm-like cells are generated by NC. Please carefully and precisely state the desired message.

Pages 7 and 8 and figure 3 panels D and E...are not clearly labeled. It is hard to understand what each image represents. Please indicate inhibitor, and length and time of treatment.

Reviewer #2 (Remarks to the Author):

The authors have thoroughly and nicely addressed all points raised by me.

This is a very important work to the field, and the potential of this much needed model to bring relevant insights into human development and disease has been nicely demonstrated.

Reviewer #3 (Remarks to the Author):

The authors successfully addressed all my comments. The paper is ready to be accepted.

REVIEWER COMMENTS (and POINT-BY-POINT RESPONSES)

Reviewer #1 (Remarks to the Author):

The reviewed manuscript incorporates recommendations and it is much improved. Really only minor suggestions from this reviewer for consideration. The Introduction, first paragraph:

Odd description of germ layers and NC contributions to the face, as no endoderm-like cells are generated by NC. Please carefully and precisely state the desired message.

Response: We appreciate your continuous review and feedback. We deleted the following sentence “Although the face consists of three germ layers,” from the first paragraph to state our message precisely. We believe this revision according to your suggestions improved our manuscript by eliminating the possibility of misleading of readers.

Pages 7 and 8 and figure 3 panels D and E...are not clearly labeled. It is hard to understand what each image represents. Please indicate inhibitor, and length and time of treatment.

Response: We apologize for inconvenience. We added labels to identify treatments for samples.

Thank you for your continuous review and comments. Your suggestions refined our manuscript, improving the accuracy of description.

Reviewer #2 (Remarks to the Author):

The authors have thoroughly and nicely addressed all points raised by me.

This is a very important work to the field, and the potential of this much needed model to bring relevant insights into human development and disease has been nicely demonstrated.

Thank you for your comments to improve our manuscript during the revision. We hope this work helped to improve the understanding of human craniofacial development and related diseases.

Reviewer #3 (Remarks to the Author):

The authors successfully addressed all my comments. The paper is ready to be accepted.

Thank you for your comments to improve our manuscript during the revision. Your suggestion expanded the potential of our *in vitro* model.

Figures and Tables:

In Fig. 3d and e, we added labels (“IWP-2” and “Dorsomorphin”) to identify treatment of samples.